

# Chemical and stable carbon isotopic compositions of PM$_{2.5}$ from two typical forests in China: Implication for sources

Mingyu Li[1], Zhanjie Xu[1], Zhichao Dong[1], Junjun Deng[1], Pingqing Fu[1], Chandra Mouli Pavuluri[1]

[1] Institute of Surface-Earth System Science, School of Earth System Science, Tianjin University, Tianjin 300072, China

*Correspondence to:* Zhanjie Xu (xuzhanjie@tju.edu.cn); Chandra Mouli Pavuluri (cmpavuluri@tju.edu.cn)

**Abstract.** To elucidate the origin and seasonality of atmospheric aerosols in forest areas, simultaneous PM$_{2.5}$ collection was carried out in two typical forest sites: Changbai Mountain (CB, 42.40N, 128.11E), North China and Xishuangbanna (BN, 22.25N, 100.89E), South China, at day and night during the summer and winter periods of 2023-2024. Carbonaceous and nitrogenous components, water-soluble inorganic ions (WSIIs) and stable carbon isotopic composition of total carbon ($\delta^{13}C_{TC}$)

were measured in PM$_{2.5}$. Generally, the contents of carbonaceous and nitrogenous components were higher in winter than summer, with secondary organic carbon (SOC) and water-soluble organic carbon (WSOC) higher in daytime than in nighttime at CB and BN. The average concentrations of WSIIs in total samples were 5.36 μg m$^{-3}$ and 2.23 μg m$^{-3}$ at CB and BN. SO$_4^{2-}$, NO$_3^-$ and NH$_4^+$ were dominant at CB, while SO$_4^{2-}$, NH$_4^+$ and Na$^+$ were dominant at BN, which accounted for 86% and 89% in BN to the total ions. $\delta^{13}C_{TC}$ ranged from −27.8‰ to −22.1‰ at CB, while −27.6‰ to −24.5‰ at BN. Based on the results

obtained, we found that besides biogenic emissions, the emissions from biomass burning and terrestrial and/or marine organisms were major sources of aerosols at both sites. Further fossil fuel combustion contributed more significantly at CB than at BN. Thus, this study provides insight into the origins and aging processes of PM$_{2.5}$ in forest areas in North and South China.

## 1 Introduction

Fine aerosols are the particulate matter with an aerodynamic diameter that does not exceed 2.5 μm (PM$_{2.5}$) in the atmosphere. The PM$_{2.5}$ can influence the Earth's climate system through direct solar radiation absorption or scattering, and indirectly by serving as cloud condensation nuclei (Liou and Ou, 1989; Ramana and Devi, 2016). In addition, PM$_{2.5}$ has been found to have harmful impacts on visibility, human health and ecosystems (Shaughnessy et al., 2015; Maji et al., 2018; Zhang et al., 2019; Xue et al., 2022; Chen et al., 2023; Zheng et al., 2024). Further the recent studies found that the PM$_{2.5}$ affects the productivity

and aggravates socio-economic inequality (Peeples, 2020; Canaday et al., 2024; Li et al., 2024).

PM$_{2.5}$ consists mainly of water-soluble inorganic ions (WSIIs), carbonaceous components and trace elements (Wang et al., 2017; Zhao et al., 2022). Among them, WSIIs account for about 20-60% of PM$_{2.5}$ and their proportion increase with increasing pollution levels (Cao et al., 2007; Tao et al., 2014; Yin et al., 2014; He et al., 2017; Guo et al., 2023). Carbonaceous components primarily consist of elemental carbon (EC) and organic carbon (OC). EC originates mainly from incomplete combustion of





biomass and fossil fuels (Sharma et al., 2022). While OC is derived from primary organic matter directly released as particulate matter from pollution sources and secondary formation from human-made or biogenic emissions of volatile organic compounds (VOCs) (Ehn et al., 2014). Notably, WSIIs promote the formation of secondary organic aerosols (SOA) (Pathak et al., 2003; Fu et al., 2024). However, tracing the sources clearly by measuring only the chemical components in aerosols is difficult. The stable carbon isotope ratio ($\delta^{13}C_{TC}$) has been proven in recent studies to aid in identifying the sources and

transformation processes of PM$_{2.5}$ (Kawamura et al., 2004; Aggarwal et al., 2013; Kunwar et al., 2016).

Currently, research on the origins of PM$_{2.5}$ has been widely carried out worldwide (Kawashima et al., 2023; Espina-Martin et al., 2024; Chen et al., 2025). Therefore, comprehensive chemical composition studies of PM$_{2.5}$ from different regions remain important. Chinese studies on atmospheric PM$_{2.5}$ are predominantly centered on large and medium-sized cities and other regions with substantial populations or severe pollution, like the Beijing-Tianjin-Hebei urban agglomeration and the Yangtze

River Delta. (Huang et al., 2014; Wang et al., 2021; Dong et al., 2023; Li et al., 2024). Nevertheless, due to the variations in pollution sources, climate, geographical location, and other factors in different background areas, the characteristics of PM$_{2.5}$ concentration and its chemical composition differ in various regions. Compared with urban areas, the composition of aerosols in forest regions is complex in natural source components, so their formation and aging mechanisms are more complicated (Bhat and Fraser, 2007; Mo et al., 2015; Ren et al., 2019; Ehn et al., 2014; Kourtchev et al., 2009). Forest plants act as major

sources of biogenic VOCs which serve as crucial precursors to PM$_{2.5}$, and can form biogenic SOA through photochemical reactions (Yuan et al., 2013; Wu et al., 2020). Furthermore, biomass burning is also a significant source of organic aerosol (OA), during which extensive biomass burning emits significant quantities of VOCs and particulate matter. These substances undergo a series of intricate chemical transformations, resulting in substantial OA formation (Long et al., 2023). This can seriously affect the air quality, inducing marked elevations in PM$_{2.5}$ mass concentrations and posing a threat to public health

(Bu et al., 2021; Chen et al., 2024; Yin et al., 2024). Therefore, research on the characterization and origin of aerosols in forest areas is necessary.

China has a vast land with a forest coverage of ca. 24% in 2022. This study selected two typical forest areas located in the south and north China for the observation and chemical analysis of PM$_{2.5}$. Here, we report the temporal variability in the concentrations and compositions of carbonaceous components, WSIIs, nitrogenous components, as well as $\delta^{13}C_{TC}$ of PM$_{2.5}$.

Based on the dataset obtained, we explore the origins and aging processes of PM$_{2.5}$ in the forest regions of north and south China.

## 2 Methodology

### 2.1 PM$_{2.5}$ Sampling

The PM$_{2.5}$ samples (*n* = 120) were collected at the Changbai Mountain Forest Ecosystem Positioning Research Station of the

Chinese Academy of Sciences (CB) in Jilin Province [42.40N, 128.11E, 740 m above sea level (asl)] and the Guanping Management Station of Xishuangbanna National Nature Reserve (BN) in Yunnan Province [22.25N, 100.89E, 872 m asl] (Fig.



1) on pre-combusted (450℃, 6 h) quartz membrane filters (405.3 cm$^2$) employing a high-volume air sampler operated at 1.0 m$^3$ min$^{-1}$ in daytime (23:30-10:30 UTC) and nighttime (11:00-24:00 UTC) in summer from 22 July to 7 August 2022 and in winter from 26 December 2022 to 9 January 2023. Prior to and after sampling, blank samples were obtained by setting the filter membrane on the sampler and allowing it to remain for 5 minutes without air pumping. The filter membrane is enclosed in aluminum foil after every sampling, and put into a sealed plastic pouch, and stored away from light. All samples preserved under -20℃ conditions until analysis.

**Figure 1: (a) Map of China with the sampling points: Changbai Mountain (CB), North China and Xishuangbanna (BN), South China, with vegetation coverage. (b, c) Clustered 72-hour backward airmass trajectories plots (above the ground level: 500m) at CB and BN, China, at summer and winter periods during 2023-24.**



## 2.2 Chemical analyses

### 2.1.1 Measurement of carbonaceous components

OC and EC were analyzed using a semi-continuous OC/EC analyzer (Sunset Laboratory, USA), which separates OC and EC
by heating the sample at different temperatures and distinguishes between OC and EC by using a laser or light source to
monitor changes in the reflected or transmitted light of the sample during the heating process. Briefly, a portion of a filter was
extracted and positioned in a quartz boat situated within the thermal desorption chamber, followed by combustion through a
two-step heating procedure.

Water-soluble OC (WSOC) was isolated from filter aliquots through ultrasonic extraction using Milli-Q water and quantified
employing a TOC analyzer (OI Analytical, model 1030W C 1088). All measured concentrations were field blank-corrected to
ensure data accuracy. The following equation was used to estimate the TC and water-insoluble OC (WIOC).

$$TC = OC + EC, \tag{1}$$

$$WIOC = OC - WSOC, \tag{2}$$

Owing to technical limitations in direct SOC determination, an EC tracer-based method was implemented for SOC assessment,
which was estimated based on the following equation (Castro et al., 1999):

$$SOC = OC - [EC \times (OC/EC)_{min}], \tag{3}$$

where $(OC/EC)_{min}$ is the minimum value of the mass concentration ratio of OC/EC produced from primary emissions.
Considering the differences in meteorological conditions and pollution source emissions at each sampling sites in different
seasons, the average value of three $(OC/EC)_{min}$ monitored in different seasons at the sampling sites was applied to estimate the
SOC. The minimum OC/EC ratios of 23.59, 7.15 in CB and 7.57, 17.26 in BN during the summer and winter, respectively.

### 2.2.2 Measurement of inorganic ions

Inorganic ion analysis was conducted using Ion chromatography (ICS 5000+, Thermo Fisher). Briefly, to measure anions, an
eluent consisting of $Na_2CO_3$, $NaHCO_3$, and $H_2SO_4$ was utilized at a controlled flow rate of 1.2 mL min$^{-1}$. For cation
determination, methyl sulfonic acid functioned as the eluent, operating at a flow rate of 1.0 mL min$^{-1}$ (Pavuluri et al., 2011a)
Generally, the error in duplicate analyses did not exceed 4%.

### 2.2.3 Measurement of nitrogenous components

Water-soluble total nitrogen (WSTN) was measured by a continuous-flow analyzer. The filter sample was ultrasonically
extracted for 10 minutes in 10 mL of MilliQ water, repeated 3 times. A PTFE membrane filter with a 0.22 µm size was used
to filter the aqueous fractions. The aqueous fractions were filtered using a 0.22 µm-sized PTFE membrane filter and
subsequently mixed with excess $K_2SO_4$. All nitrogen (N) is converted to nitrate ($NO_3^-$) through ultraviolet digestion and then



reduced to nitrite ($NO_2^-$). After that, the $NO_2^-$ reacts with aminobenzene sulfonic acid to produce new nitrogen compounds. Using an UV spectrophotometer to measure the sample solution's absorbance at 540 nm.

The inorganic nitrogen (IN) concentration was calculated by aggregating the measured concentrations of $NO_2^-$-, $NO_3^-$-, and $NH_4^+$-N. The concentration difference between WSTN and IN was regarded as water-soluble organic nitrogen (WSON) (Pavuluri et al., 2015; Dong et al., 2023).

$$[IN] = [NO_3^--N] + [NH_4^+-N] + [NO_2^--N], \tag{4}$$

$$[WSON] = [WSTN] - [IN], \tag{5}$$

**2.2.4 Determination of stable carbon isotope ratios of TC**

Stable carbon isotope ratio of TC ($\delta^{13}C_{TC}$) in $PM_{2.5}$ was analyzed through a Flash 2000HT elemental analyzer connected to a 253 Plus isotope ratio mass spectrometer (EA-IRMS). Overall, an aliquot of the filter was wrapped and injected into EA, with the evolved gases $CO_2$ delivered to an IRMS *via* ConFlo-II for the determination of $^{13}C/^{12}C$ in TC. The delta ($\delta$) values represent the isotope ratio of $^{13}C/^{12}C$, in parts per million (ppm) with reference to Pee Dee Belemnite for carbon. The isotope conversion equation is as follows:

$$\delta^{13}C_{TC} = [(^{13}C/^{12}C)_{sample} / (^{13}C/^{12}C)_{standard} - 1] \times 1000, \tag{6}$$

The error derived from replicate analysis remains within 0.3‰. The samples were not decarbonized before measurement. Due to the fact that average $Ca^{2+}$ concentrations were found to be low (0.03 ± 0.02 μg m⁻³, CB; 0.09 ± 0.07 μg m⁻³, BN) in these samples so that we assume that the contribution of $CaCO_3$ and its $\delta^{13}C$ to aerosols is negligible (Wang et al., 2005; Pavuluri et al., 2011b).

**2.3 Meteorological parameters and simulations of air mass trajectories**

Meteorological parameters, including temperature, relative humidity, and wind speed at CB and BN, were from the Xihe Energy Big Data Platform (https://xihe-energy.com/#geo). To analyze air mass transport patterns, 72-hour backward trajectories arriving at CB and BN at an altitude of 500 m above the ground level were calculated at 6-hour intervals employing the Hybrid Single Particle Lagrangian Integrated Trajectory (HYSPLIT) model from the National Oceanic and Atmospheric Administration (https://www.ready.noaa.gov/index.php).

**3 Results and discussion**

**3.1 Differences in meteorology and long-range transported air masses between the sampling sites**

Figure 1 illustrates 72-hour backward air mass trajectory clusters. It revealed the air masses that reaching CB originated primarily from the Pacific Ocean, including the Yellow Sea (68%), the Sea of Japan (24%) and the East China Sea (8%), while





BN was primarily affected by the southwestern airflow from the Indian Ocean from the Bay of Bengal, as well as airflows

passing through the Indo-China Peninsula in summer. In winter, the air masses to CB were derived from Siberia (51.0%), North China (30.0%) and Mongolia (9.0%). Compared with CB, the local source contribution at BN during winter was higher. The temporal variations of meteorological parameters at two sites are depicted in Fig. 2. The ambient temperatures at CB and BN during the campaigns exhibited seasonal variations. The temperatures at BN were similar to those at CB during summer, whereas in winter, the temperatures at BN (avg. 14.8°C) were significantly higher than those at CB (avg. -10.9°C). Both the

pressure and wind speed were found to be higher at CB. Significant diurnal variations in relative humidity were observed at both the sites, where CB demonstrated daytime and nighttime values of 84.84% and 72.03% respectively, compared to BN's corresponding measurements of 90.84% and 73.96%.

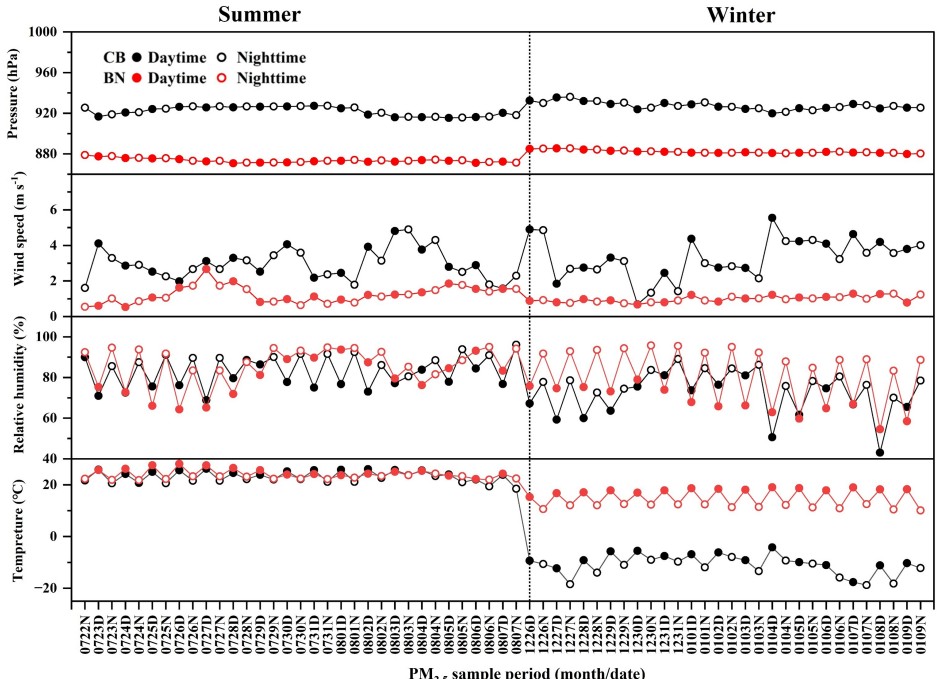

**Figure 2: Temporal variations of meteorological parameters at CB and BN, China during 2023-24.**

**3.2 Chemical results**

Table 1 shows concentrations of carbonaceous (EC, OC, SOC, WSOC and WIOC), nitrogenous (WSTN, IN and WSON), WSIIs and $\delta^{13}C_{TC}$ in $PM_{2.5}$ at CB and BN, China in this study.

**Table 1: Annual and seasonal summary of concentrations of carbonaceous (EC, OC, SOC, WSOC and WIOC), nitrogenous (WSTN, IN and WSON) components and WSIIs (Cl⁻, $NO_3^-$, $SO_4^{2-}$, Na⁺, K⁺, $NH_4^+$, $Ca^{2+}$ and $Mg^{2+}$) (μg m⁻³) and $\delta^{13}C_{TC}$ in $PM_{2.5}$ at CB and**

**BN, China during 2023-24.**

| Components | Annual | Summer | Winter |
|---|---|---|---|




| | Range/med | Avg ± SD | Range/med | Avg ± SD | Range/med | Avg ± SD |
|---|---|---|---|---|---|---|
| | CB (*n* = 60) | | CB (*n* = 31, Jul-Aug) | | CB (*n* = 29, Dec-Jan) | |
| | BN (*n* = 60) | | BN (*n* = 30, Jul-Aug) | | BN (*n* = 30, Dec-Jan)) | |
| **Carbonaceous components ($\mu g\ m^{-3}$)** | | | | | | |
| EC | n.d. - 0.68/0.16 | 0.17 ± 0.14 | n.d. - 0.09/0.05 | 0.02 ± 0.03 | n.d. - 0.68/0.19 | 0.14 ± 0.22 |
| | n.d. - 0.41/0.19 | 0.19 ± 0.09 | n.d. - 0.41/0.11 | 0.16 ± 0.11 | 0.13 - 0.34/0.23 | 0.23 ± 0.05 |
| OC | 0.74 - 9.64/2.27 | 2.73 ± 1.72 | 0.74 - 3.51/1.56 | 1.67 ± 0.67 | 1.72 - 9.64/3.34 | 3.87 ± 1.78 |
| | 1.22 - 6.75/3.87 | 3.75 ± 1.33 | 1.22 - 6.75/2.71 | 2.93 ± 1.26 | 2.80 - 5.90/4.73 | 4.58 ± 0.77 |
| WSOC | 0.41 - 5.97/1.19 | 1.48 ± 1.00 | 0.41 - 1.83/0.87 | 0.95 ± 0.40 | 0.80 - 5.97/1.73 | 2.04 ± 1.14 |
| | 0.42 - 5.84/2.41 | 2.16 ± 1.03 | 0.42 - 5.84/1.30 | 1.55 ± 1.10 | 1.63 - 3.45/2.81 | 2.77 ± 0.45 |
| WIOC | 0.81 - 3.87/1.57 | 1.83 ± 0.77 | 0.33 - 1.68/0.60 | 0.71 ± 0.34 | 0.81 – 3.87/1.57 | 1.83 ± 0.77 |
| | n.d. – 2.76/1.71 | 1.80 ± 0.44 | n.d. - 3.00/1.35 | 1.44 ± 0.56 | 1.15 - 2.76/1.71 | 1.80 ± 0.44 |
| SOC | n.d. - 6.77/1.57 | 1.74 ± 1.14 | n.d. - 2.27/1.15 | 1.16 ± 0.60 | n.d. - 6.77/1.97 | 2.36 ± 1.28 |
| | n.d. - 5.62/1.03 | 1.34 ± 1.09 | n.d. - 5.62/1.82 | 1.98 ± 1.20 | n.d. - 1.50/0.81 | 0.73 ± 0.42 |
| WSOC/OC | 0.34 - 0.72/0.51 | 0.51 ± 0.09 | 0.42 - 0.80/0.55 | 0.57 ± 0.09 | 0.34 - 0.72/0.52 | 0.52 ± 0.09 |
| | 0.47 - 0.69/0.62 | 0.61 ± 0.05 | 0.25 - 1.04/0.48 | 0.50 ± 0.15 | 0.47 - 0.69/0.62 | 0.61 ± 0.05 |
| OC/EC | n.d. - 82.07/22.04 | 26.92 ± 16.42 | 23.03 - 82.07/36.67 | 43.94 ± 19.69 | 6.15 - 29.15/19.39 | 19.29 ± 5.35 |
| | 6.82 - 52.31/21.34 | 22.90 ± 9.78 | n.d. - 52.31/25.54 | 25.41 ± 13.42 | 15.02 - 25.15/20.74 | 20.56 ± 2.32 |
| SOC/OC | n.d. – 0.99/0.63 | 0.58 ± 0.19 | n.d. – 0.99/0.59 | 0.50 ± 0.29 | n.d. - 0.75/0.63 | 0.61 ± 0.11 |
| | n.d. – 3.45/2.05 | 1.78 ± 1.13 | n.d. – 0.86/0.72 | 0.63 ± 0.20 | 1.63 – 3.45/2.81 | 2.77 ± 0.45 |
| WIOC/OC | 0.28 - 0.66/0.49 | 0.49 ± 0.09 | 0.20 - 0.58/0.45 | 0.43 ± 0.09 | 0.28 - 0.66/0.49 | 0.47 ± 0.09 |
| | n.d. - 0.53/0.38 | 0.39 ± 0.05 | n.d. - 0.75/0.52 | 0.52 ± 0.12 | 0.31 - 0.53/0.38 | 0.39 ± 0.05 |
| **Nitrogenous components ($\mu g\ m^{-3}$)** | | | | | | |
| WSTN | 0.01 – 11.89/1.54 | 2.33 ± 2.34 | n.d. – 3.06/0.91 | 0.97 ± 0.84 | 0.87 – 11.89/3.48 | 3.80 ± 2.54 |
| | n.d. – 2.57/1.21 | 1.29 ± 0.51 | n.d. – 3.25/0.32 | 0.47 ± 0.74 | 0.63 – 2.57/1.21 | 1.29 ± 0.51 |
| IN | 0.44 – 5.89/1.65 | 1.85 ± 1.26 | 0.02- 1.40/0.41 | 0.53 ± 0.35 | 0.44 – 5.89/1.65 | 1.85 ± 1.26 |
| | n.d. - 1.10/0.48 | 0.52 ± 0.23 | n.d. - 0.42/0.13 | 0.15 ± 0.11 | 0.19 – 1.10/0.48 | 0.52 ± 0.23 |
| WSON | n.d. – 5.99/0.90 | 1.32 ± 1.22 | n.d. – 1.90/0.42 | 0.58 ± 0.51 | 0.43 –5.99/1.78 | 1.92 ± 1.29 |
| | n.d. – 1.47/0.71 | 0.77 ± 0.30 | n.d. – 2.93/0.32 | 0.58 ± 0.85 | 0.31 – 1.47/0.71 | 0.77 ± 0.30 |
| **Water – soluble inorganic ions ($\mu g\ m^{-3}$)** | | | | | | |
| $Cl^-$ | n.d. - 0.52/0.02 | 0.06 ± 0.09 | n.d. - 0.06/0.01 | 0.02 ± 0.01 | n.d. - 0.52/0.06 | 0.10 ± 0.11 |
| | n.d.- 0.12/0.02 | 0.03 ± 0.02 | n.d. - 0.12/0.02 | 0.02 ± 0.03 | n.d. - 0.07/0.03 | 0.04 ± 0.02 |
| $SO_4^{2-}$ | 0.17 - 9.27/1.78 | 2.31 ± 1.71 | 0.17 - 5.84/1.46 | 1.89 ± 1.38 | 0.89 - 9.27/2.11 | 2.74 ± 1.91 |
| | 0.04 - 3.61/0.96 | 1.07 ± 0.74 | 0.04 - 1.31/0.53 | 0.60 ± 0.35 | 0.65 - 3.61/1.39 | 1.54 ± 0.72 |
| $NO_3^-$ | 0.01 - 8.71/0.08 | 1.10 ± 1.79 | 0.01 - 0.08/0.02 | 0.03 ± 0.02 | 0.23 - 8.71/1.46 | 2.19 ± 2.03 |
| | n.d. - 0.44/0.04 | 0.07 ± 0.07 | n.d. - 0.08/0.04 | 0.04 ± 0.02 | 0.02 - 0.44/0.05 | 0.09 ± 0.10 |
| $Na^+$ | n.d. - 2.66/0.35 | 0.56 ± 0.48 | n.d. - 2.66/0.20 | 0.29 ± 0.45 | 0.23 - 1.46/0.84 | 0.83 ± 0.32 |
| | n.d. – 0.72/0.32 | 0.33 ± 0.19 | 0.03 – 0.72/0.31 | 0.30 ± 0.17 | n.d. - 0.72/0.34 | 0.39 ± 0.21 |
| $NH_4^+$ | 0.03 - 5.04/0.93 | 1.18 ± 0.95 | 0.03 - 1.79/0.52 | 0.67 ± 0.45 | 0.50 - 5.04/1.42 | 1.71 ± 1.04 |
| | n.d. - 1.40/0.32 | 0.41 ± 0.33 | n.d. - 0.53/0.16 | 0.17 ± 0.14 | 0.24 - 1.40/0.57 | 0.64 ± 0.30 |
| $K^+$ | n.d. - 0.68/0.08 | 0.13 ± 0.14 | n.d. - 0.09/0.03 | 0.03 ± 0.02 | 0.08 - 0.68/0.18 | 0.22 ± 0.14 |
| | n.d. - 0.50/0.16 | 0.17 ± 0.12 | n.d. - 0.41/0.07 | 0.10 ± 0.09 | n.d. - 0.50/0.25 | 0.24 ± 0.09 |
| $Mg^{2+}$ | n.d. - 0.13/0.03 | 0.04 ± 0.03 | n.d. - 0.13/0.04 | 0.05 ± 0.03 | n.d. - 0.04/0.01 | 0.01 ± 0.01 |
| | n.d. - 0.02/0.01 | 0.01 ± 0.01 | n.d. - 0.02/0.07 | 0.01 ± 0.00 | n.d. - 0.02/0.01 | 0.01 ± 0.01 |
| $Ca^{2+}$ | n.d. - 0.07/0.03 | 0.03 ± 0.02 | n.d. | | n.d. - 0.07/0.03 | 0.03 ± 0.02 |
| | n.d. - 0.28/0.07 | 0.09 ± 0.07 | | | n.d. - 0.28/0.07 | 0.09 ± 0.07 |
| **Isotope ratios (‰)** | | | | | | |
| $\delta^{13}C_{TC}$ | (27.8) – (22.1)/(25.9) | (25.7) ± 1.5 | (27.8) – (26.2)/(27.0) | (27.0) ± 0.5 | (26.2) – (22.1)/(24.6) | (24.6) ± 1.1 |
| | (27.6) – (24.5)/(25.8) | (26.0) ± 0.9 | (27.6) – (26.2)/(26.8) | (26.9) ± 0.4 | (25.9) – (24.5)/(25.3) | (25.3) ± 0.4 |



### 3.2.1 Characterization of inorganic ions and nitrogenous components

The linear regressions of total cations and anions were shown in Fig. 3. The ratio of the equivalent concentrations of cation (CE) and anion (AE) can effectively evaluate the acid-base balance of aerosols. The formulas are as follows:

$$AE = \frac{Cl^-}{35.5} + \frac{SO_4^{2-}}{48} + \frac{NO_3^-}{62}, \tag{7}$$


$$CE = \frac{Na^+}{23} + \frac{NH_4^+}{18} + \frac{K^+}{39} + \frac{Mg^{2+}}{12} + \frac{Ca^{2+}}{20}, \tag{8}$$

If the CE is greater than the AE, PM$_{2.5}$ is alkaline, and *vice versa*. Average annual equivalent ratios of total cations (Na$^+$, NH$_4^+$, K$^+$, Mg$^{2+}$ and Ca$^{2+}$) to anions (Cl$^-$, NO$_3^-$ and SO$_4^{2-}$) were 1.62 ± 0.53 µg m$^{-3}$ in CB and 1.92 ± 0.80 µg m$^{-3}$ in BN, indicating that the aerosols at these two sites are alkaline. The high ratio of AE/CE might be due to the enhanced NH$_3$ emission caused by high temperature and agricultural activities (Qiao et al., 2019). It also should be noted that the excessive cations in both

sites might be related to unmeasured anions like carbonate and oxalate. Furthermore, soluble organic acid ions might also be the cause of the anion deficits in CB and BN. A more comprehensive investigation of this matter will be conducted in future research initiatives.

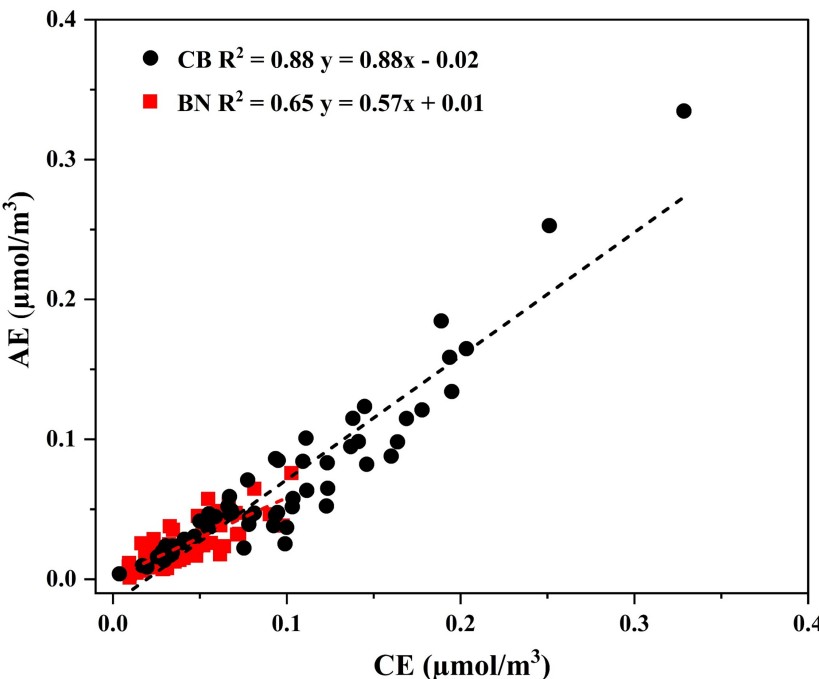

**Figure 3: Anion and cation equilibrium in PM$_{2.5}$ collected from CB and BN.**

The concentration statistics of WSIIs in CB and BN PM$_{2.5}$ samples were shown in Table 1. On average, SO$_4^{2-}$ was identified as the predominant ionic species at both sites (CB: 2.31 µg m$^{-3}$; BN: 1.07 µg m$^{-3}$). They accounted for 43% and of total ionic mass at CB and 52% at BN, respectively. NH$_4^+$ was the second most abundant ion (CB: 1.18µg m$^{-3}$; BN: 0.41µg m$^{-3}$) followed



by $NO_3^-$, $Na^+$, $K^+$, $Cl^-$, $Mg^{2+}$ and $Ca^{2+}$ at CB, whereas at BN, their abundances followed an order: $Na^+ > K^+ > NO_3^- > Ca^{2+} > Cl^- >$ $Mg^{2+}$. As the main secondary ions in PM$_{2.5}$, the cumulative concentration $SO_4^{2-}$, $NO_3^-$ and $NH_4^+$ reached 86% at CB and 76%

at BN of the total ions, respectively. Interestingly, $NH_4^+$ and $Na^+$ were the second and third abundant ions, in total ions at BN. The concentrations of main secondary ions were significantly lower compared to those typically observed in urban sites, such as Tianjin, Beijing, Guangzhou, Chongqing in China, Chennai in India, and Hachinohe in Japan (Pathak et al., 2009; Qiao et al., 2019; Pavuluri et al., 2011a; Dong et al., 2023; Sun and Zhang, 2024). Except for $Na^+$ and $K^+$, The mean levels of different ions at CB exceeded those at BN. Ionic species exhibited peak concentrations in winter in their seasonal distributions (Fig. 4).

However, the ionic species did not show a clear diurnal variation.

Coal combustion is a major source of $SO_2$, $NO_x$, and $NH_3$ (Zhang et al., 2020; Zheng et al., 2022). The concentrations of $SO_4^{2-}$, $NH_4^+$, and $NO_3^-$ in PM$_{2.5}$ during winter were higher than those in summer, being 1.45, 2.55, and 73.00 times, respectively, at CB, and 2.57, 3.76, and 2.25 times, respectively, at BN. Moreover, the concentration of secondary ions at CB were higher than that at BN in winter. This could be linked to the increased utilization of coal for domestic heating in winter,

which leads to substantial emissions of gaseous precursors like $SO_2$ and $NO_x$. Additionally, the low temperatures in winter resulted in a decreased atmospheric boundary layer, which hindered the dispersion of pollutants. The consistently higher concentrations of $SO_4^{2-}$ at CB might be associated with CB's proximity to the coast. Oceanic phytoplankton and/or dimethyl sulfide (DMS) emitted from biomass burning undergo photochemical oxidation to convert into $SO_2$, which subsequently transformed into $SO_4^{2-}$.

Figure 5 summarizes the concentrations and percentage contributions of various water-soluble ionic components. The contribution of $NO_3^-$ to total ions at BN was very small in both summer (3%) and winter (3%), which may be attributed to the removal effect of wet deposition on nitrate particles due to the hot and humid climate throughout the year. However, the lower temperature environment (<15°C) in winter might facilitated the transformation of gaseous nitric (HNO$_3$) acid to particulate (NH$_4$NO$_3$), thereby potentially increasing the concentration of particulate $NO_3^-$. On the other hand, anthropogenic activities

such as winter heating might emit more NOx, which, after undergoing a sequence of chemical reactions in the atmosphere, were converted into $NO_3^-$. Therefore, the contribution of $NO_3^-$ (28%) in winter at CB was relatively high. In addition, the higher contribution of $Na^+$ (25%) and the very low contribution of $Mg^{2+}$ (about 0%) at BN during summer implies that the $Na^+$ could have been originated primarily from marine emissions rather than from soil dust.

The average concentration of WSTN was $2.33 \pm 2.34$ µg m$^{-3}$, and IN was $1.85 \pm 1.26$ µg m$^{-3}$ at CB, while $1.29 \pm 0.51$ µg m$^{-3}$

and $0.52 \pm 0.23$ µg m$^{-3}$ at BN. WSTN, IN, WSON and secondary ions had the same seasonal variation trend in concentration, with higher levels in winter (Fig. 6). WSON constituted an average of 57.7% of WSTN at CB and 40.3% at BN., respectively. WSON/WSTN at CB and BN were significantly higher as than other sites located in Himalaya (18%, hill site), New Delhi, India (19%, urban site), Sapporo, Japan (9%, urban site) and Svalbard Islands, Norway (8%, Coastal site) (Pavuluri et al., 2015; Tripathee et al., 2021; Boreddy et al., 2024; Pei et al., 2024). Similarities were noted with the forest aerosols collected from

Rondônia, Brazil, during an intense biomass burning period. (~45%, forest site) (Mace et al., 2003).



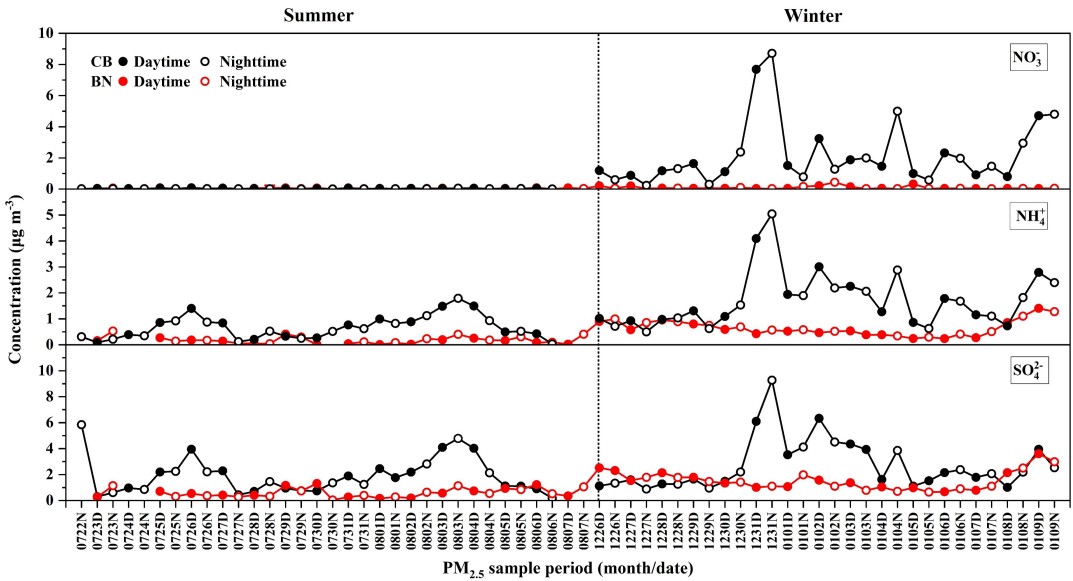

**Figure 4: Temporal variations of secondary ionic species concentrations (µg m⁻³) in PM₂.₅ collected from CB and BN, China during 2023-24.**

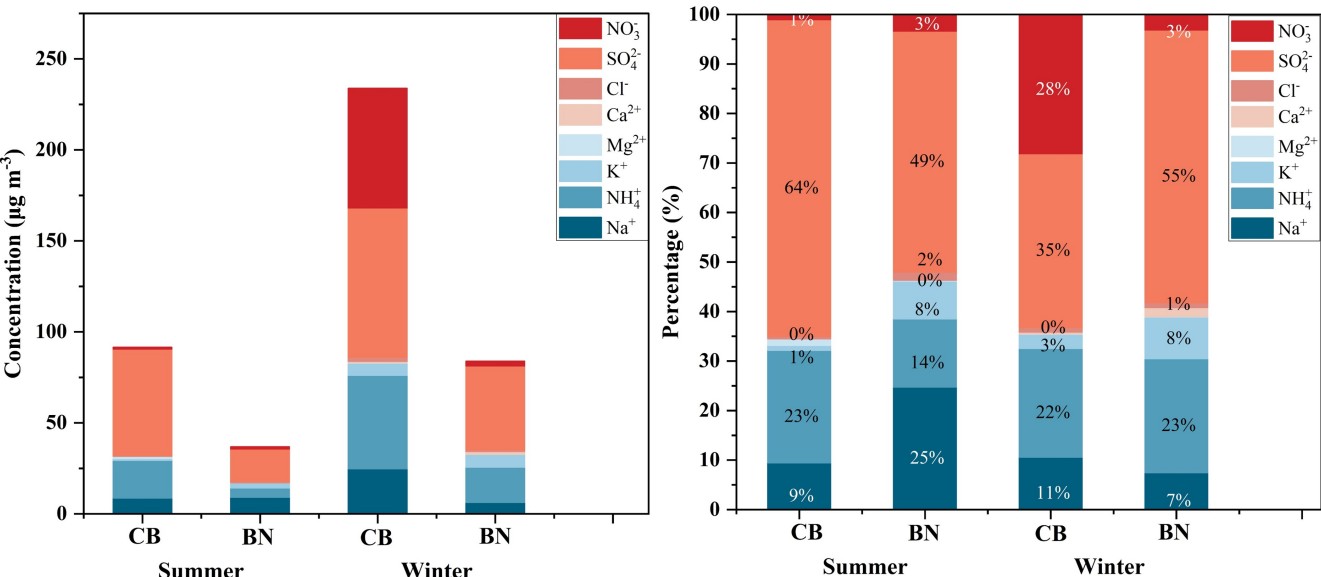

**Figure 5: Concentrations and percentages of WSIIs in total ions.**



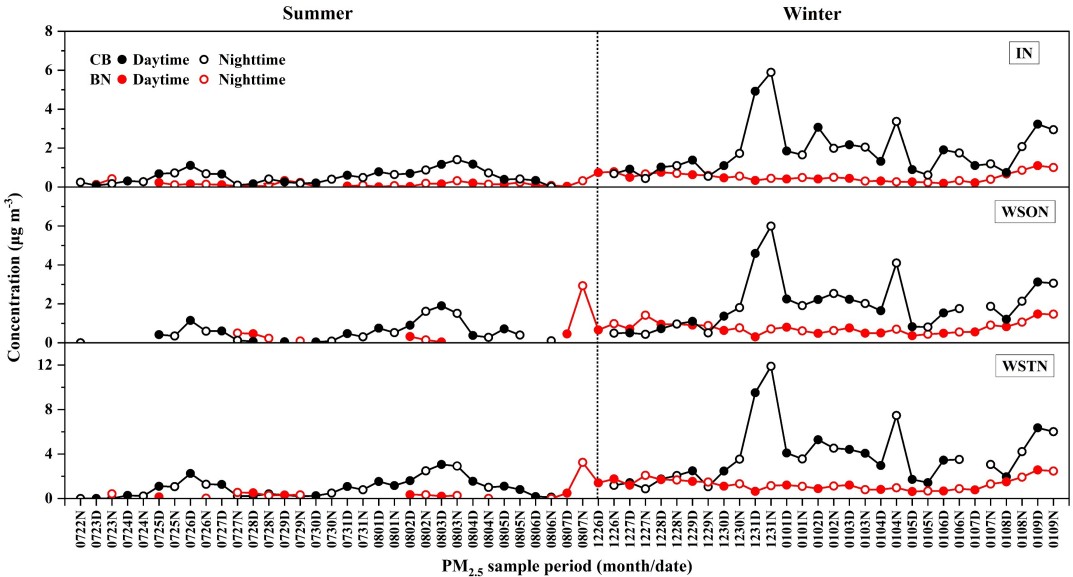

**Figure 6: Temporal variations of concentrations (µg m$^{-3}$) of nitrogenous components in PM$_{2.5}$ collected from CB and BN, China during 2023-24.**

### 3.2.2 Characterization of carbonaceous components and $\delta^{13}C_{TC}$

Table 1 and Figure 7 show the mass concentrations and temporal variations of OC, EC, SOC, and WSOC in PM$_{2.5}$ at CB and BN in China. The average mass concentrations of OC and EC at CB were 2.73 ± 1.72 µg m$^{-3}$ and 0.17 ± 0.14 µg m$^{-3}$, while those at BN were 3.75 ± 1.33 µg m$^{-3}$ and 0.19 ± 0.09 µg m$^{-3}$. Furthermore, average concentrations of OC were consistently higher than EC at both sites. OC, WSOC and SOC exhibited notable seasonal variations (Fig. 7), and their average concentrations were all higher in winter compared to summer. Overall, OC levels in winter were twice as high as in summer

at both CB and BN. EC in winter were 7 times higher at CB but only 1.4 times higher at BN compared to that in summer. Elevated EC levels in winter suggested a higher influence of fossil fuel combustion. Moreover, the higher loads of OC in contrast to EC in winter and summer at both sites imply that secondary OC formation and/or increased emissions from coal combustion and biomass burning were significant. The average concentration of SOC at CB in winter (2.36 ± 1.28 µg m$^{-3}$) was twice as high as in summer (1.16 ± 0.60 µg m$^{-3}$). In addition, the average concentration of WSOC was 1.46 ± 1.00 at CB

and 2.16 ± 1.03 µg m$^{-3}$ at BN. BN exhibits a higher level of WSOC, suggesting that there might be higher emissions and/or more secondary formation occurred under conditions of greater oxidant abundance at BN than at CB. Moreover, the temporal variations of OC, WIOC, SOC, and WSOC exhibited comparable trends, suggesting a common or similar source origin and potentially similar formation processes at CB and BN.

The concentrations of OC, SOC and WSOC were higher during the daytime than at nighttime at CB and BN (Fig. 9). The

intense sunlight and high temperatures prevalent in the local region might have facilitated the enhanced formation of SOC during the summer. However, EC displayed no significant diurnal variation at CB. As shown in Fig. 7, OC, SOC and WSOC




at CB in winter showed similar diurnal trends, suggesting that they could share similar/same origins and formation processes. Elevated concentrations of carbonaceous components at CB on December 31, 2023, was observed, which could be attributed to local fireworks and firecracker celebrations in advance of the New Year's Eve. We also noticed an increase in the concentrations of OC, SOC, and WSOC in the daytime on July 30th and July 25th, 2023, at BN, which might be related to local biomass burning events.

Figure 9 illustrates the seasonal and annual variations in $\delta^{13}C_{TC}$ of CB and BN aerosols. The annual $\delta^{13}C_{TC}$ variability in $PM_{2.5}$ ranged from -27.8‰ to -22.1‰, (avg. -25.7 ± 1.5‰) at CB, while -27.6 to -24.5‰ (avg. -26.0 ± 0.9‰) at BN during the campaign. Overall, the $\delta^{13}C_{TC}$ at CB was more positive than that at BN. Compared with winter, the $\delta^{13}C_{TC}$ in summer were much lower, whose values ranged from -27.8 to -26.2‰ (avg. -27.0 ± 0.5‰) at CB, and from -27.6 to -26.2‰ (avg. -26.9 ± 0.4‰) at BN.

The diurnal variation of $\delta^{13}C_{TC}$ in the aerosols of CB and BN was not very significant, except for a slight difference in winter at CB and in summer at BN, where the average values were -24.8 ± 1.0‰ and -27.1 ± 0.3‰ at the daytime, -24.5 ± 1.2‰ and -26.7 ± 0.3‰ at the nighttime, respectively. Their values were more positive at nighttime. This might be associated with stronger plant emissions/biological activity during the daytime, and higher humidity and lower temperatures at nighttime that favored gas-to-particle transformation of organic compounds (Ren et al., 2019).

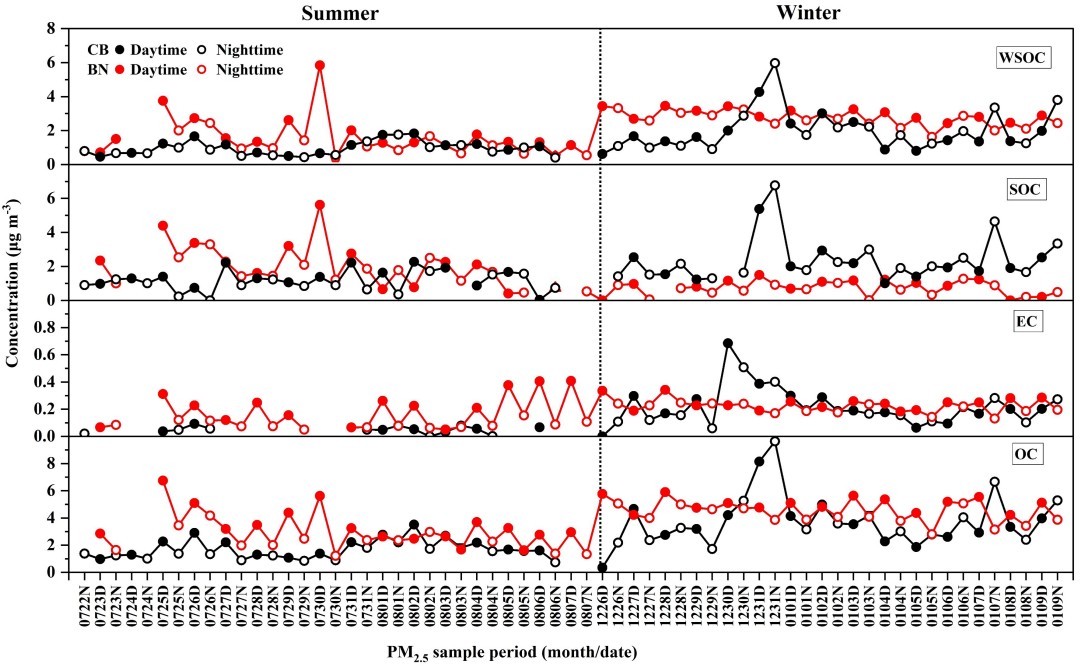

**Figure 7: Temporal variations in the concentrations of OC, EC, WSOC, and SOC in $PM_{2.5}$ collected from CB and BN, China during 2023-24.**



**Figure 8: Diurnal variations in carbonaceous components in summer and winter forest aerosol samples collected from CB and BN, China during 2023-24.**






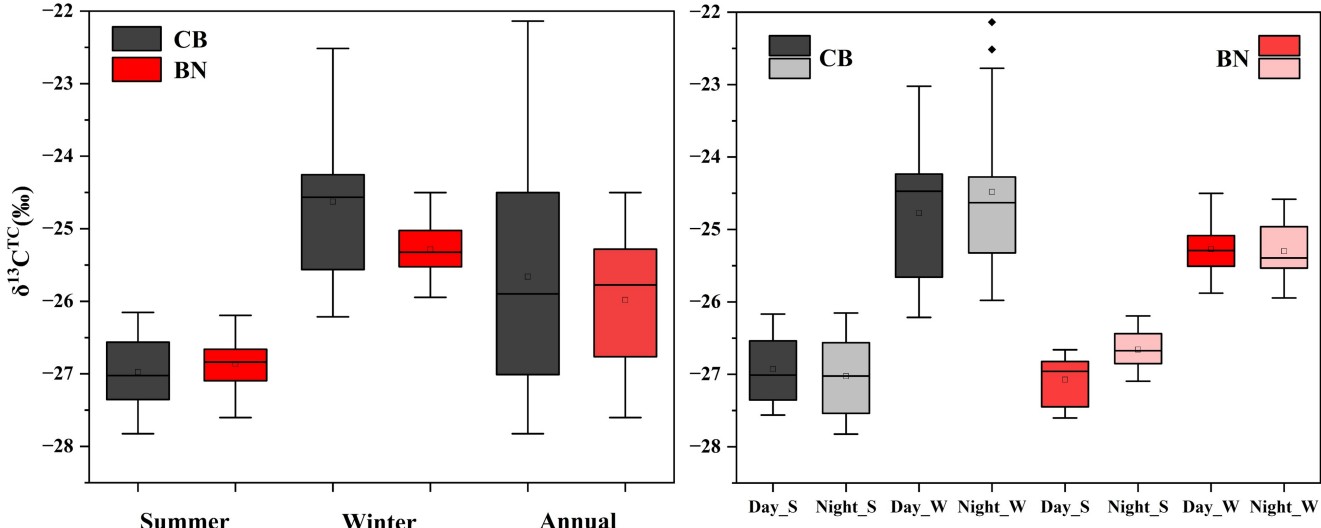

**Figure 9: Seasonal and diurnal variations of $\delta^{13}C_{TC}$ collected form CB and BN during the campaign. The S and W represent summer and winter, respectively.**

### 3.3 Implications for PM$_{2.5}$ sources

### 3.3.1 Origins of inorganic ions

Characterization of the ionic components of aerosols from primary sources can reflect the changing characteristics of the emission sources. nss-K$^+$ serves as a tracer for emissions from biomass combustion, Cl$^-$ is associated with emissions from combustion activities (coal and straw burning, etc.) and Ca$^{2+}$ is from crustal sources. As shown in Fig. 10, Cl$^-$ and Mg$^{2+}$ exhibited a relatively strong correlation (summer: $R^2 = 0.63$; winter: $R^2 = 0.51$), and had a weak correlation with Na$^+$ (summer: $R^2 = 0.43$) and NO$_3^-$ (summer: $R^2 = 0.32$), indicating that Cl$^-$ was also influenced by natural sources such as soil dust and marine sources. Ca$^{2+}$, as a tracer of soil and dust sources, exhibited a characteristic concentration trend with lower levels in summer and higher levels in winter at BN. It might be attributed to the scavenging effect of frequent precipitation in summer on soil and dust. Mg$^{2+}$ exhibited a positive correlation with Ca$^{2+}$ ($R^2 = 0.72$) during winter at BN, suggesting that Mg$^{2+}$ might be derived from soil dust. The elevated levels of Na$^+$ at BN during summer were driven by marine air masses originating from the Bay of Bengal and Pacific Ocean, transported *via* southwest and southeast air currents.

The contributions of nss-SO$_4^{2-}$ and nss-K$^+$ to the total SO$_4^{2-}$ and K$^+$ concentrations were on average 87 % (summer: 78%; winter: 96%) and 85 % (summer: 76%; winter: 94%) in BN, 92 % (summer: 95%; winter: 88%) and 76 % (summer: 69%; winter: 83%) at CB, indicating a predominant influence of anthropogenic sources over marine sources, especially in winter. Since the anthropogenic sources at CB contributed more to the ions in PM$_{2.5}$, the concentration of total ions at CB was consistently higher than that at BN in both seasons, with the annual average of total ion concentration at CB being approximately 2.63 times to that at BN. There was a significant correlation among SO$_4^{2-}$, NH$_4^+$, and NO$_3^-$ at CB (Fig. 10), implying a common origin



and analogous oxidation processes. The positive correlation between nss-$K^+$ and $NO_3^-$, $NH_4^+$, $SO_4^{2-}$ at CB ($R^2 > 0.50$) implies

that they were predominantly influenced by biomass burning.

The concentrations of secondary inorganic ions are mainly associated with the content of their gaseous precursors ($SO_2$, $NO_x$, $NH_3$) and the chemical reactions they undergo in the atmosphere, as well as meteorological conditions. The precursor of $NO_3^-$ primarily originates from vehicle exhaust emissions, while the precursor of $SO_4^{2-}$ mainly stems from coal combustion emissions from industries, residential life, and other sources. Thus, the mass ratio of $NO_3^-/SO_4^{2-}$ can evaluate the influence of

mobile and stationary sources on atmospheric aerosols. The $NO_3^-/SO_4^{2-}$ in winter is higher than in summer, yet the average ratio remains below 1, indicating that emissions from stationary sources contribute more than those from mobile sources. However, the $NO_3^-/SO_4^{2-}$ at CB site in winter ($0.79 \pm 0.39$) was approximately 26 times that in summer ($0.03 \pm 0.02$), indicating a notable rise in the contribution of mobile sources during winter. Annual average temperature of BN was about 20°C, with a relatively large volatilization of $NO_3^-$, leading to a lower $NO_3^-/SO_4^{2-}$ value (avg. $0.08 \pm 0.08$). The lower ratio in summer at both

sites, combined with the backward trajectories of air masses suggest that the air masses from ocean, rich in emissions from marine biogenic sources, might lead to a significant contribution of biogenic $SO_4^{2-}$ in summer. Moreover, $Na^+$ and $SO_4^{2-}$ at both BN ($R^2 = 0.62$) and CB ($R^2 = 0.57$) showed a moderate correlation in summer, further proving that a portion of $SO_4^{2-}$ has come from the ocean.

The molar ratio of $NH_4^+/SO_4^{2-}$ in aerosols can be used to determine the acidity of aerosols and the combination form of the main

secondary inorganic ions (Pathak et al., 2004; Lyu et al., 2015). When $1.5<NH_4^+/SO_4^{2-} \le 2$, the aerosols are almost completely neutralized although ammonium nitrate is present. If $NH_4^+/SO_4^{2-}< 1.5$, the concentration of free acid in the particulate phase is relatively high, and there is almost no $NH_4NO_3$ present. The average ratio of $NH_4^+/SO_4^{2-}$ at CB was 1.74 (summer: 0.97; winter: 2.54), whereas that at BN was 0.84 (summer: 0.44; winter: 1.25), suggesting an excess of ammonium existed in the aerosols at both sites, and nearly no $NH_4NO_3$ was present at BN. In fact, the concentrations of $NO_3^-$ and $Cl^-$ were extremely low at both

sites in summer, so the main form of $NH_4^+$ were $(NH_4)_2SO_4$ and $NH_4HSO_4$. This resulted from elevated summer temperatures enhancing the breakdown of particulate $NH_4NO_3$ into gaseous $NH_3$ and $HNO_3$. $SO_4^{2-}$ was strongly correlated with $NH_4^+$ at both sites, particularly in winter, which further demonstrates that $(NH_4)_2SO_4$ and $NH_4HSO_4$ were their primary forms. However, a favorable correlation existed between $NH_4^+$ and $NO_3^-$ at CB, and the fitting slope of ammonium and $2[SO_4^{2-}] + [NO_3^-]$ in winter was greater than 1(Fig.11), indicating that there was an adequate amount of $NH_3$ in the atmosphere to undergo neutralization

reactions with $H_2SO_4$ and $HNO_3$, thereby forming ammonium $NH_4NO_3$ and $(NH_4)_2SO_4$.



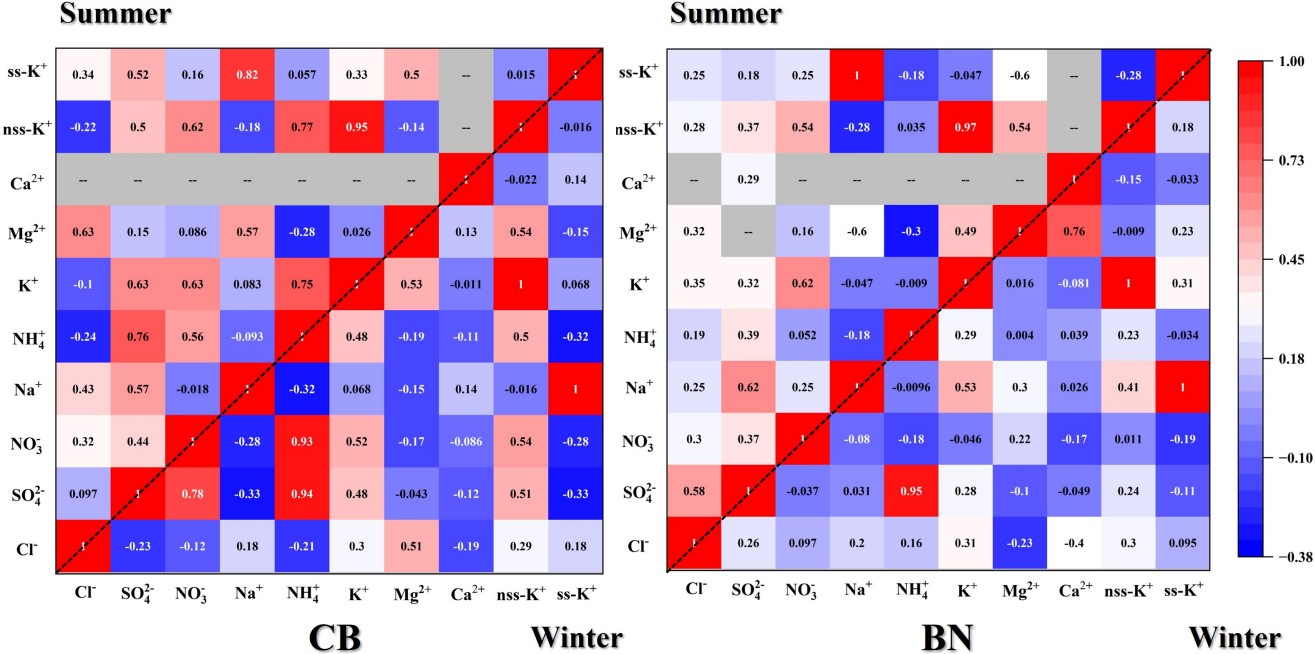

**Figure 10: Correlation heatmap of Pearson correlation coefficients of water-soluble ions.**

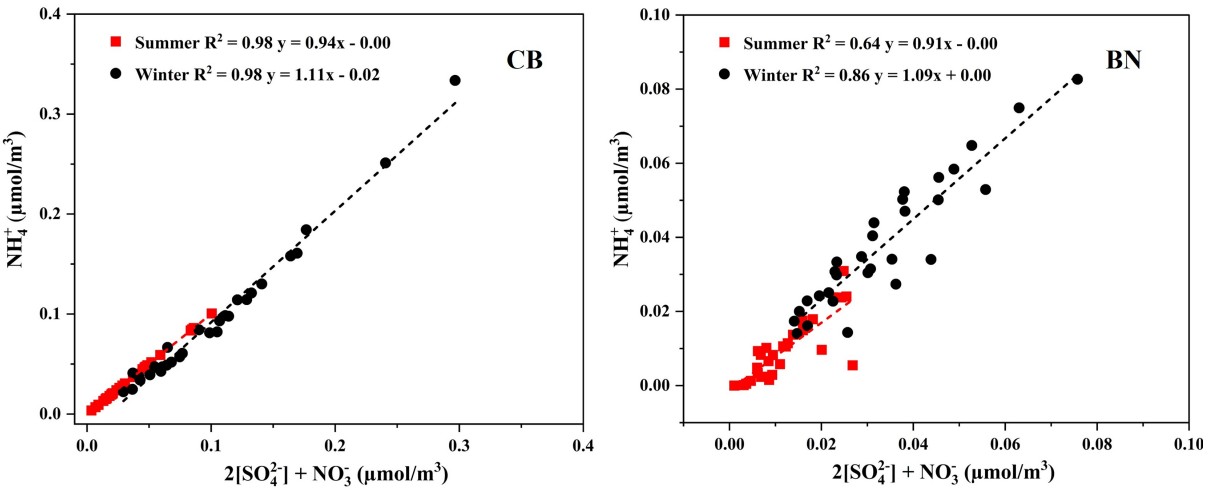

**Figure 11: Linear correlations of secondary ions in PM$_{2.5}$ collected from CB and BN during the campaign period (2023-2024).**

### 3.3.2 Origins of carbonaceous components

The variation in the correlation coefficient between OC and EC was primarily affected by pollutant sources, meteorological conditions and seasonal changes. A good correlation between OC and EC suggested a single emission source, whereas a weak correlation indicated a more complex emission source. Consequently, investigating the correlation between OC and EC across



different seasons can help infer the sources of carbonaceous aerosols. There were moderate correlations between OC and EC
during winter at CB ($R^2$ = 0.46) and BN ($R^2$ = 0.67), but weak correlations in summer at both sites (Fig. 12). This implied that
OC and EC likely shared similar sources in winter but diverged in summer.

Research indicates that in the regression equation OC = aEC + b, the term *aEC* reflects primary OC emissions from
combustion sources (e.g., coal and traffic), while *b* represents OC from non-combustion sources (Cao et al., 2007). The
value of a at CB and BN were notably higher in winter in contrast to that in summer, implying a prominent contribution from
combustion-related emissions during the winter season. The higher b value at BN in summer, implying that the impact of non-
combustion source emissions was greater in summer than in winter.

The OC/EC ratios differ among various pollution sources, so the OC/EC is useful for identifying the sources and emission
characteristics of carbonaceous components in $PM_{2.5}$. When the OC/EC > 2.0, it can be inferred that the secondary formation
of OC is likely present. Studies have shown that OC/EC ratios ranging from 1.0 to 4.2 indicate traffic sources, with
approximately 0.8 for heavy diesel vehicles and 2.2 for light gasoline vehicles; ratios of 2.5 to 10.5 imply coal combustion;
ratios of 16.8 to 40 indicate biomass burning. Average OC/EC ratios were 26.92 ± 16.42 at CB and 22.90 ± 9.78 at BN, which
closely matched the values reported for biomass burning emissions. Table 2 showed the concentrations of OC and EC in $PM_{2.5}$
in urban and forest areas. It was not difficult to find that OC/EC in forest areas was considerably higher than that in urban
areas. The average OC/EC were 13.5 and 11.5, significantly exceeding 2.0, indicating substantial SOA formation, with higher
ratios observed in summer. The elevated OC levels in summer likely resulted from the typically higher temperatures and
stronger solar radiation promoted the active life activities in forest areas, which favored the emission of VOCs and thus promote
the occurrence of photochemical reactions.

**Table 2: Comparison of mass concentrations of OC, EC (μg m$^{-3}$) and OC/EC for forest aerosols over the world.**

| City/nation | Samling period | OC | EC | OC/EC | Reference |
|---|---|---|---|---|---|
| Look Rock, US | 30 Jun.–14 Aug., 2001 | 5.6 | 0.66 | 8.5 | (Tanner et al., 2004) |
| Duke Forest, US | 10–23 Jul. 2003 | 3.2 | 0.2 | 16 | (Bhat and Fraser, 2007) |
| K-pusta, Hungary | 4 Jun.–10 Jul. 2003 | 4.00 | 0.21 | 19 | (Kourtchev et al., 2009) |
| Changbai Mountain, China | Jul. 2007 | 4.9 | 0.5 | 9.8 | |
| Chongming Island, China | Jun. 2006 | 9.9 | 1.6 | 6.2 | (Li et al., 2010) |
| Dinghu Mountain, China | Aug. 2006 | 5.3 | 0.7 | 7.6 | |
| Jiangfengling, Hainan, China | Nov. 2007 | 2.4 | 0.2 | 12 | |
| Hyytiälä, Finland | Jun. – Aug. 2007 | 1.2 ± 0.7 | 0.10 ± 0.06 | 12 | (Aurela et al., 2011) |
| | Dec. 2007-Jan. 2008. | 1.3 ± 1.2 | 0.24 ± 0.19 | 5.42 | |
| Mt. Hua, China | Jan. 2009 Jul. -Aug. 2009 | 6.0 ± 2.5 | 0.8 ± 0.5 | 7.5 | (Meng et al., 2014) |
| Baimaquan, China | 18–30 Jul. 2010 | 15.86 | 1.75 | 9.1 | |
| Panzhihua, China | 18–30 Jul. 2010 | 20.81 | 5.97 | 3.5 | (Mo et al., 2015) |
| Gongga Mountain, China | 17–31 Jul. 2011 | 3.11 | 0.42 | 7.4 | |
| Wolong, China | 16 Jul. to 2 Aug., 2012 | 9.33 | 1.42 | 6.6 | |
| Mt. Wuyi, China | 2014-2015 | 1.6 ± 0.86 | 0.48 ± 0.20 | 3.22 | (Ren et al., 2019) |
| | | 4.6 ± 1.90 | 0.69 ± 0.13 | 5.26 | |
| Olympic Forest Park, China | 2014winter | 49.17 ± 15.3 | 7.82 ± 4.07 | 6.29 | (Chen et al., 2020) |
| CB, China | 2023-2024 | 2.73 ± 1.72 | 0.17 ± 0.14 | 26.92±16.42 | This study |
| BN, China | | 3.75 ± 1.33 | 0.19 ± 0.09 | 22.90±9.78 | |



WSOC can be derived directly from biomass burning or form through atmospheric oxidation of VOCs (Schnelle-Kreis et al., 2007; Tang et al., 2020). When biomass burning influence is minimal, the WSOC/OC is regarded as an indicator of photochemical aging during long-range atmospheric transport. The average WSOC/OC was $0.51 \pm 0.09$ at CB and $0.61 \pm 0.11$ at BN. These results suggested that WSOC constituted a significant fraction of OC. Their range and average at CB and BN (Table 1) were comparable to those reported at urban sites, Tianjin, China (range 0.37–0.84, avg. 0.63) (Wang et al., 2018), Chennai, India (range 0.23–0.6; avg. 0.45) (Pavuluri et al., 2011a), Mt. Tai, China (0.55) (Fu et al., 2012), Gwangju, Korera (range 0.26 – 0.73, average 0.52) (Cho and Park, 2013) and Chengdu, China (avg. 0.50) (Tao et al., 2013), where biomass burning was regarded as the primary aerosol source, undergoing aging. In fact, during summer, higher temperatures and stronger solar radiation result in more vigorous plant activity in forest areas, leading to increased emissions of VOCs. Additionally, both regions were affected by air masses which originated from the ocean during summer. These oceanic air masses were enriched with emissions from marine organisms, which subsequently undergo photochemical oxidation during long-range atmospheric transport. Consequently, high WSOC/OC at both sites during summer were likely driven by SOA formation, linked to increased $O_3$ levels, solar radiation, and VOC emissions (Xiang et al., 2017). A strong correlation between WSOC and OC was found at CB ($R^2 = 0.84$, summer; $R^2 = 0.83$, winter) and BN ($R^2 = 0.77$, summer; $R^2 = 0.75$, winter), indicating that WSOC and OC share similar sources across different seasons.

SOC at CB and BN accounted for 65.2% and 38.4% of OC, respectively, and its proportions were higher in summer. The elevated SOC levels in summer likely resulted from increased atmospheric photochemical oxidation. The low SOC/OC ratio (0.15) indicated that in winter at BN, the OC was mainly directly driven by local primary emissions (e.g., biomass burning, coal burning, and transportation), rather than by photochemical secondary transformation. The backward trajectory of the air mass in winter at BN further confirms that. However, SOC/OC at CB (0.57) in winter was about four times that at BN. It has been shown that biomass burning significantly increases emissions of both primary and secondary aerosols (Fu et al., 2012; Zheng et al., 2018). Specifically, these activities can enhance the formation of SOA by increasing emissions of compounds such as monoterpenes and through the action of the resultant $O_3$ and NOx. Thus, the high SOC/OC at CB during winter might be linked to the effective promotion of secondary aerosol formation by burning wood for heating. Furthermore, the daytime and nighttime samples collected at CB in winter exhibited diurnal variations in SOC concentrations, further confirming the significant contributions of local anthropogenic emissions and photochemical oxidation.

The WIOC fraction in the samples likely contains substantial quantities of partially combusted biogenic residues. It was noteworthy that SOC exhibited a moderate correlation with WIOC at CB ($R^2 = 0.69$; $R^2 = 0.38$) but showed no correlation at BN ($R^2 = 0.08$; $R^2 = 0.03$; Fig. 12). This implied that the source of WIOC and SOC might be similar at CB, with a significant portion of SOC being water-insoluble, while primary emissions contributed substantially at BN. The average WIOC/OC ratios at CB (0.43, summer: 0.43; winter: 0.47) and BN (summer: 0.50; winter: 0.39) were comparable to that (0.55) reported in Chennai, India, which were regarded as predominantly originating from biomass burning and undergoing aging during long-range transport (Pavuluri et al., 2011a).





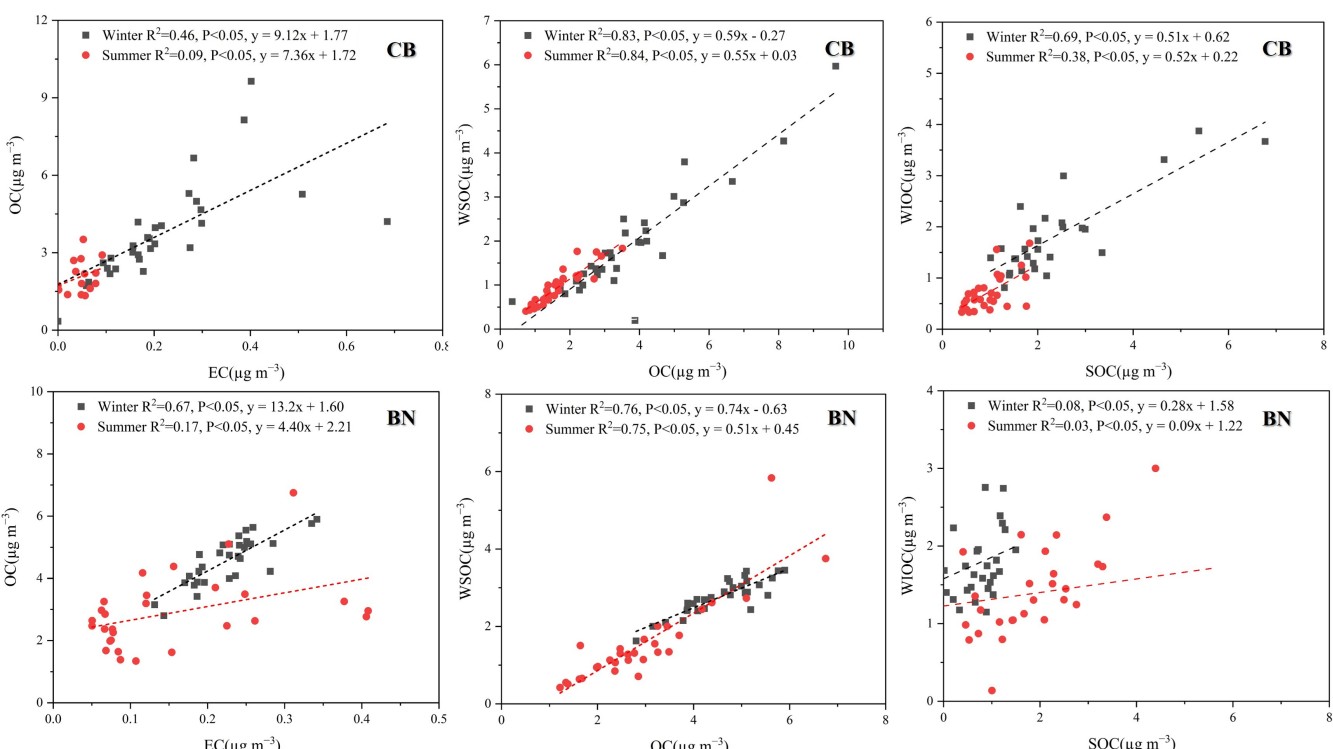

**Figure 12: Correlations of certain carbonaceous components in PM$_{2.5}$ collected from CB and BN, China during 2023-24.**

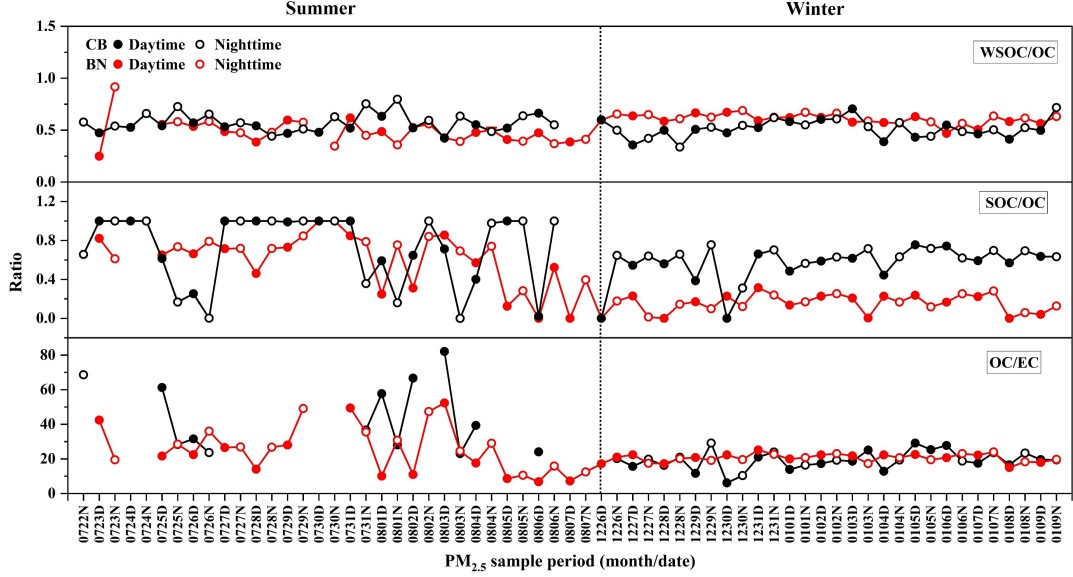

**Figure 13: Temporal variations of OC/EC, WSOC/OC and SOC/OC in PM$_{2.5}$ collected from CB and BN, China during 2023-24.**



### 3.3.3 Impact of biomass burning based on $\delta^{13}C_{TC}$

The $\delta^{13}C_{TC}$ observed at CB and BN during summer were comparable to those reported for aerosols in Singapore, Indonesia (-27.5 to -26.0‰) and Delhi, India (-27.7 to -24.9‰), where it was clearly indicated that the emissions from the burning of $C_3$ plants were the primary source of aerosols (Narukawa et al., 1999; Agarwal et al., 2024). The $\delta^{13}C$ value of aerosol derived

from fossil fuel combustion was significantly higher compared to those originating from biomass burning. Moreover, Boreddy et al. (2018) had observed depletion of $^{13}C$ in aerosols linked to increased contributions from $C_3$ plant burning in Southeast Aisa and reported that photochemical aging of aerosols could have led to an increase in $\delta^{13}C_{TC}$ values (Boreddy et al., 2018). The $\delta^{13}C_{TC}$ ranged from -26.2 to -22.1‰ at CB (avg. -24.6 ± 1.1‰) and ranged from -25.9 to -24.5 at BN (avg. -25.3 ± 0.4‰) in winter could be linked to the consumption of fossil fuels for heating and the aging of aerosols transported from continental

and marine regions.

However, no significant correlations were found between WSOC/OC and $\delta^{13}C_{TC}$ for the biomass burning aerosols, suggesting that OA at CB and BN were mostly derived from primary emissions (Cao et al., 2016). It has shown that the average values of $\delta^{13}C_{TC}$ in the remote marine aerosols ranged from -24.0 to -18.1‰ (Kawamura et al., 2017; Verwega et al., 2021). In this research, the $\delta^{13}C_{TC}$ ranged from -27.8‰ to -22.1‰ at CB and from -27.6‰ to -24.5 at BN, indicating that the contribution of

marine aerosols during the campaign was relatively small. The values of $\delta^{13}C_{TC}$ in CB and BN aerosols might be the result of multiple sources' contribution, such as fossil fuel combustion (coal, natural gas and petroleum) (-28 to -21‰), marine phytoplankton (-28 to -15‰) and $C_3$ plants biomass burning (-28 to -26‰) (Singh et al., 2018). The concentration of $SO_4^{2-}$ and $NO_3^-$ at CB increased in winter, so the elevated values of $\delta^{13}C_{TC}$ were likely related to fossil fuel burning for heating. Therefore, fossil fuel combustion and $C_3$ plant biomass burning are identified as primary sources, while also being influenced by air

masses transported from the ocean during the summer.

### 4 Conclusions

Day- and night-time $PM_{2.5}$ samples ($n$ = 120) were collected at two typical forest areas in North and South China. Their carbonaceous, nitrogenous, WSIIs components and $\delta^{13}C_{TC}$ have been measured. The concentrations of carbonaceous and nitrogenous components displayed a distinct seasonal trend, higher in winter and lower in summer. In addition, the OC, SOC

and WSOC showed diurnal variations in which their concentrations were higher during daytime than nighttime. The relationships and mass ratios of carbonaceous components indicated that the $PM_{2.5}$ at CB and BN were dominantly from biomass burning and photochemical reactions of VOCs. A high proportion of $Na^+$ was found at BN in summer, which in combination with the backward trajectory of the air mass suggested the influence of oceanic air masses. We also found significant deficiencies of anions in aerosols from CB and BN. Moreover, correlations between nss-$K^+$ and secondary ions

suggested that the aerosols at two forest sites were affected by biomass burning and the primary source emissions were more important. The values of $\delta^{13}C_{TC}$ reaffirmed that biomass burning was the primary source of $PM_{2.5}$ at two sites, while they were also influenced by fossil fuel burning during winter, especially at CB, and additionally by oceanic air masses. Regarding the



anion deficiency phenomenon found in the aerosols at the two sites in this study, we will conduct further analysis in combination with organic acids in the subsequent research to better indicate the sources and composition characteristics of
aerosols in forest regions.

**Author contributions:**

*ML & ZX: Conceptualization, Investigation, Methodology, Writing–original draft. ZD: Investigation, Writing – review & editing; JD: Writing – review & editing. PF: Writing – review & editing. CMP: Funding acquisition, Writing – review &*
*editing. ZX: Project administration, Funding acquisition, Supervision, Writing – review & editing.*

**Competing interests:**

*The authors declare that they have no conflict of interest.*

**Acknowledgements:**

*This study is supported by the National Natural Science Foundation of China (Nos. 42277090 and 42202199). We appreciate*
*Rui Liu and Jin Liu for their assistance in the field investigation.*

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
