# Peer review of "Chemical and stable carbon isotopic compositions of PM2.5 from two typical forests in China: Implication for sources"

_EGUsphere, 2025_

## Author Comment (AC1)

RC1: 'Comment on egusphere-2025-1335', Anonymous Referee #1, 26 May 2025

**Authors' Response to Referee #1 Comments**

Dear reviewer,

Thank you very much for your critical reading of the manuscript, appreciation of our work and comments/suggestions, which helped to improve the quality of the manuscript. The manuscript is revised accordingly, and our point-by-point responses to all the comments are provided below. Please see the revised manuscript for details of the revisions.

1. Writing and Readability

The manuscript is generally well-structured encompassing the conventional format of scientific articles. The language is clear and concise, facilitating comprehension. However, there are occasional grammatical errors and awkward phrasings that could benefit from careful proofreading. For instance, the sentence:

"Further fossil fuel combustion contributed more significantly at CB than at BN." could be rephrased for clarity as:

"Furthermore, fossil fuel combustion contributed more significantly at CB than at BN."

Overall, the manuscript is accessible to readers with a background in atmospheric sciences or environmental chemistry.

*Response:* Thank you for your insightful comments and suggestions. We have revised the sentence as recommended. We have further carefully reviewed and revised the manuscript. In response, we have:

- Page 1, Line 25: Further the recent studies found that the $PM_{2.5}$ affects the productivity and aggravates socio-economic inequality.
  *We have modified it to:* Furthermore, recent studies have found that $PM_{2.5}$ affects the productivity and aggravates socio-economic inequality.
- Page 2, Line 37: Currently, research on the origins of $PM_{2.5}$ has been widely carried out worldwide
  *We have modified it to:* Recent research on the sources of $PM_{2.5}$ has been extensively conducted worldwide.
- Page 3, Line 73:The filter membrane was enclosed in aluminum foil after every sampling, and put into a sealed plastic pouch, and stored away from light.
  *We have modified it to:* The filter membrane was immediately wrapped in aluminum foil, sealed in a plastic pouch, and stored away from light after sampling.
- Page 4, Line 80: OC and EC were analyzed using a semi-continuous OC/EC analyzer (Sunset Laboratory, USA), which separates OC and EC by heating the sample at different temperatures and distinguishes between OC and EC by using a laser or light

source to monitor changes in the reflected or transmitted light of the sample during the heating process.

*We have modified it to:* The mass concentrations of OC and EC were measured using a semi-continuous thermal/optical OC/EC analyzer (Sunset Laboratory, USA). The distinction between OC and EC is achieved via real-time monitoring of light reflectance/transmittance changes during the heating process using a laser/light source, i.e. the IMPROVE protocol of the protective visual environment (Wan et al., 2017; 2015).

2. Scientific Content

2.1. Experimental Design

The study investigates the chemical and stable carbon isotopic compositions of $PM_{2.5}$ collected from two forest sites in China—Changbai Mountain (CB) in the north and Xishuangbanna (BN) in the south—during summer and winter periods. The sampling strategy includes day and night collections, providing temporal resolution. The analysis encompasses carbonaceous and nitrogenous components, water-soluble inorganic ions (WSIIs), and $\delta^{13}C$ of total carbon ($\delta^{13}C_{TC}$).

While the study design is comprehensive, there are concerns regarding certain methodological choices:

Filter Material: The use of quartz filters for collecting $PM_{2.5}$ samples intended for WSII analysis is questionable. Quartz filters are known to have high blank values for certain ions, which can interfere with accurate quantification. Although the authors mention using blanks to correct for background levels, the inherent high background of quartz filters, especially for cations like $Na^+$, $Ca^{2+}$, and $Mg^{2+}$, can compromise the reliability of the measurements. Alternative filter materials, such as Teflon or PCT, are more suitable for WSII analysis due to their lower blank values.

*Response:* Thank you for raising this important point regarding filter selection for WSII analysis. We agree that quartz filters may exhibit higher blanks for certain ions. In our study, quartz filters were selected primarily for their high collection efficiency for $PM_{2.5}$ and thermal stability, and compatibility with multi-component analysis (including fatty acids, aldehydes/ketones). To minimize random variability, field blanks were collected both at the beginning and end of sampling for each site and season, followed by blank correction. Currently, quartz filters are also widely used for ion analysis in numerous studies (Meng et al., 2014; Cao et al., 2017).

However, after analysis, it was found that the concentrations of the samples from the Mt. Changbai site were higher than Xishuangbanna. Moreover, the blank values in Mt. Changbai site were generally low (The $Ca^{2+}$, and $Mg^{2+}$ in the blank were almost 0). The interference of the blank values of the quartz filters on the samples could be ignored through blank correction. For the samples from the Xishuangbanna site, the inherently high background of the quartz filters did interfere with $Na^+$, $Ca^{2+}$, and $Mg^{2+}$ in the samples, which was also the reason for the ion imbalance at Xishuangbanna. Therefore, during the analysis, we tried our best to avoid analyzing these cations from the Xishuangbanna site. We have provided an explanation in Section 2.2.2 (Page4, Line106) and made corresponding

revisions to the analysis in Section 3.3.1 (Page15, Line 268-280). Please refer to the revised manuscript for detailed modifications.

*References:*

*Cao, F., Zhang, S.-C., Kawamura, K., and Zhang, Y.-L.: Inorganic markers, carbonaceous components and stable carbon isotope from biomass burning aerosols in Northeast China, Science of The Total Environment, 572, 1244-1251, https://doi.org/https://doi.org/10.1016/j.scitotenv.2015.09.099, 2016.*

*Meng, J., Wang, G., Li, J., Cheng, C., Ren, Y., Huang, Y., Cheng, Y., Cao, J., and Zhang, T.: Seasonal characteristics of oxalic acid and related SOA in the free troposphere of Mt. Hua, central China: Implications for sources and formation mechanisms, Science of The Total Environment, 493, 1088-1097, https://doi.org/10.1016/j.scitotenv.2014.04.086, 2014.*

Trajectory Analysis: The study employs backward trajectory analysis at a fixed altitude of 500 meters to infer the potential sources of air masses. However, this approach may not accurately represent the transport pathways of surface-level aerosols, especially considering the diurnal variation of the planetary boundary layer (PBL). During nighttime, the PBL can be shallow, and air masses at 500 meters may reside in the residual layer, not interacting with surface emissions. Therefore, trajectory analyses should consider the dynamic nature of the PBL and possibly include multiple altitudes to capture a more representative range of transport pathways.

*Response:* We agree with your insights. Through literature review, we found that the daytime PBL height in the study area during summer can reach 1000-1200 m, while the nocturnal stable boundary layer may shrink to 200–400 m (Wu et al., 2024). Based on this, we have recalculated 72-hour backward trajectories for three characteristic heights (300 m, 500 m, and 1000 m) during typical sampling days to more comprehensively characterize aerosol sources. We corrected the descriptions in Section 3.1 (Page 7, Line 143). In addition, we also removed the air mass trajectory in Figure 1 (Page 3) and placed the new air mass trajectory in Figure 2 (Page 7) in the revised MS.

*Reference:*

*Wu Wenlu, Chen Haisha, Guo Jianping, Xu Zhiqi and Zhang Xiaoyan: Regionalization of the Boundary-Layer Height and its Dominant Influence Factors in Summer over China [J]. Chinese Journal of Atmospheric Sciences (in Chinese), 48, 1201-1216, https://doi.org/10.3878/j.issn.1006-9895.2212.22183, 2024.*

[Figure]

Figure 3: Clustered 72-hour backward airmass trajectories plots (above the ground level: 300m, 500m, and 1000m) at Mt. Changbai and Xishuangbanna, China during 2023-24.

2.2. Data Presentation and Interpretation

The results indicate seasonal and diurnal variations in PM$_{2.5}$ composition, with higher concentrations of carbonaceous and nitrogenous components in winter. The dominance of SO$_4^{2-}$, NO$_3^-$, and NH$_4^+$ at CB, and SO$_4^{2-}$, NH$_4^+$, and Na$^+$ at BN, is reported. The $\delta^{13}$C$_{TC}$ values suggest contributions from biomass burning and fossil fuel combustion.

While the data presentation is generally clear, there are areas where further clarification is needed:

Ion Balance: The study does not discuss the ion balance between measured cations and anions. An imbalance could indicate missing species or analytical errors. Given the use

of quartz filters, which can introduce artifacts, a discussion on ion balance would strengthen the reliability of the WSII data.

*Response:* For the samples from the Xishuangbanna site, the inherently high background of the quartz filters did interfere with $Na^+$, $Ca^{2+}$, and $Mg^{2+}$ in the samples, which was also the reason for the ion imbalance at Xishuangbanna. Therefore, during the analysis, we tried our best to avoid analyzing these cations from the Xishuangbanna site. We added an explanation in Section 2.2.2 in the revised MS (Page4, Line 106).

Source Apportionment: The authors rely on $\delta^{13}C_{TC}$ values and the relative abundance of chemical species to infer sources. However, more robust source apportionment techniques, such as Positive Matrix Factorization (PMF) or Chemical Mass Balance (CMB) modeling, could provide quantitative estimates of source contributions and reduce uncertainty.

*Response:* Thank you for your suggestions. We conducted an analysis in combination with PMF modeling. After preliminary evaluation, the current data exhibits the following issues, which may not meet the requirements for the PMF model.
We attempted to run the analysis on samples from Mt. Changbai with relatively higher concentrations than Xishuangbanna, but found that the dataset contains missing values, and some key species (such as $Cl^-$, EC, $NO_3^-$, and $K^+$) have concentrations below the detection limit. There are a total of 100 pieces of $\delta^{13}C_{TC}$ data, some of which were not measured due to their too low concentrations. Therefore, there is also the problem of many missing values. These results in an incomplete data matrix, which may compromise the stability of the model.
Additionally, the low-concentration data has a poor signal-to-noise ratio (SNR), and the high measurement uncertainty makes the PMF analysis susceptible to random errors.
Finally, the interpretability of the model's factors is weak. After multiple test runs, we found that the resolved factor profiles poorly match the characteristics of potential pollution sources, making it difficult to identify their origins based on environmental significance or literature comparisons (e.g., $K^+$ and $Cl^-$, conventional biomarkers for biomass burning, exhibited no significant common factor).

2.3. Novelty and Contribution

The study contributes to the limited data on $PM_{2.5}$ composition in forested regions of China, particularly regarding $\delta^{13}C_{TC}$ measurements. However, the findings largely corroborate existing knowledge about the sources and seasonal variations of $PM_{2.5}$. The use of $\delta^{13}C_{TC}$ as a tracer is valuable, but its application here does not yield novel insights into source apportionment beyond what is already known.

Specific Comments

Line 117: The authors state that the contribution of $CaCO_3$ to aerosols is negligible, yet later identify soil as a significant source of $PM_{2.5}$. This appears contradictory, as soil dust typically contains substantial amounts of calcium carbonate. Clarification is needed to reconcile these statements.

*Response:* We have corrected this statement in combination with the problem of ion measurement (Page15, 268-279).

Table 1: The table1 appears to have two different legends, which may cause confusion. Ensuring consistency in table legends is essential for clarity.

*Response:* Thank you for pointing out this important issue. Following the Referee #2 suggestion. We have moved Table 1 to the Supplementary Material (Table S1).

1. Conclusion

The manuscript provides valuable data on $PM_{2.5}$ composition in two forested regions of China, with a focus on seasonal and diurnal variations. However, methodological concerns, particularly regarding filter selection for WSII analysis and the trajectory analysis approach, need to be addressed. The study's findings align with existing literature, and while the inclusion of $\delta^{13}CTC$ measurements is commendable, it does not substantially advance the understanding of $PM_{2.5}$ sources.

Recommendation: Major revisions are necessary to address the methodological issues and enhance the robustness of the source apportionment analysis. Incorporating more suitable filter materials, refining trajectory analyses, and employing quantitative source apportionment models would significantly strengthen the study.

*Re:* Thank you again for your detailed and constructive review. We have revised the MS as recommended. We sincerely hope that these revisions are satisfactory and get your approval for final publication.

---

## Author Comment (AC2)

*RC2: 'Comment on egusphere-2025-1335', Anonymous Referee #2, 18 Jun 2025*

**Authors' Response to Referee #2 Comments**

This article presents a measurement report of the concentration of main inorganic ions, of EC/ OC, and carbon isotope in fine particulate matter ($PM_{2.5}$) collected at two sites in China. The introduction section is clear and I sincerely appreciate the references up to date. Nevertheless, references are lacking in the other sections to assess the methodology and the discussion of the results. The dataset is interesting and deserves publication. Nevertheless, I have some major concerns regarding the methodology, the results and the discussion.

We thank the referee very much for his/her critical reading of the manuscript, appreciation of our work and the comments/suggestions, which helped to improve the quality of the manuscript further. The MS has been revised accordingly, and the point-by-point responses to each of the comments and suggestions are provided below.

- The two sites presented should be better described. Many research stations are available in China. Why is it interesting to compare these two in particular?

  *Response: For question 1,* we have added detailed description for the two sites. Changbai Mountain (temperate monsoon climate, coniferous - deciduous mixed forest) and Xishuangbanna (tropical monsoon climate, tropical rainforest) represent typical forest sites in the temperate and tropical zones of China respectively. The former is dominated by the northwest monsoon in winter, with significant aerosol transport characteristics; the latter has a highly natural environmental background value. Due to the differences in vegetation and climate between the two sites, their regulatory mechanisms for aerosol emission, deposition and composition are different. Please see Lines 60-68 in Section 2.1 in the revised MS.
  *For question 2,* Comparing the chemical and stable carbon isotopic compositions characteristics of $PM_{2.5}$ in these two sites is conducive to a better understanding of the influence mechanisms of different latitude forest ecosystems on atmospheric composition, and also provides a certain reference for the verification of different atmospheric models.

- Lines 74-78: please add a reference for the methodology used for EC/OC measurement

  *Response:* We have added two references (i.e., Wan et al. 2015; 2017) for the methodology used for EC/OC measurement in the revised MS (see Line 83).

- Lines 92-95: the description of the analytical methodology is not sufficient.

  *Response:* We have supplemented and improved the detailed description of the analytical methodology in the revised MS (see Line 98-110).

- Lines 98-99: repetition of the description of the filtration step.

  *Response:* We removed part of the description of the filtration step in the revised MS to avoid the repetition in the revised MS (see Line 113).

- Lines 101-102: the analytical methodology for the detection of $NO_2^-$ is not clear: $NO_2^-$ reacts with the acid to form "new nitrogen compounds", which are detected at 540 nm… which compounds?

  *Response:* The $NO_2^-$ reacts with aminobenzene sulfonic acid to produce high molecular weight nitrogen compounds (azo dye). Using an UV spectrophotometer to measure the total N's absorbance at 540 nm. We have added this point in the revised MS (see Lines 115-117).

- Line 103: "aggregating" means "sum"?

  *Response:* Yes, we have replaced "aggregating" with "summing" for more precise expression in the revised MS (see Line 118).

- Line 117: The authors assume that carbonates are negligible since $Ca^{2+}$ concentrations are low. But carbonates can be also as $Na^+$ or $Mg^{2+}$ salts. A measurement of inorganic carbon would be more appropriate.

  *Response:* We agree with the reviewer's opinion that the existence of $Na_2CO_3$ and $MgCO_3$ is likely and the measurement of inorganic carbon would be more appropriate, if the dust contribution is highly significant. In fact, the concentrations of both $Ca^{2+}$ and $Mg^{2+}$ were found to be extremely low at both the sites (see Table S1 in the revised MS). Furthermore, we found significant positive correlations between $Na^+$ and $SO_4^{2-}$ and $Cl^-$ at both the sites (CB: r = 0.43 with $SO_4^{2-}$, 0.57 with $Cl^-$; BN: r = 0.25 with $SO_4^{2-}$, 0.62 with $Cl^-$). Marine-derived $Na^+$ primarily exists as NaCl, $Na_2SO_4$, and $NaNO_3$. While coal combustion and biomass burning contribute to $Na^+$ emissions, typically in the forms of NaCl, $Na_2SO_4$, and $NaNO_3$. This is particularly evident at CB in winter, where increased coal burning elevates $Na^+$ concentrations. Therefore, we assume that the carbonate level in these forest samples is negligible and strongly believe that this assumption would not affect the obtained results and the drawn conclusions.

- Figure 2: why one point par day instead of a continuous temporal variation?

  *Response:* We collected meteorological data and calculated the average values for day and night, which were consistent with the collection cycle of each of our samples. This was for the convenience of analysis.

- Paragraph 3 (results and discussion): The two sites are at low altitude, respectively CB 740 m asl and BN 872 m asl. I assume that they are in the boundary layer in summer, but CB could be in the free troposphere in winter. A discussion should be added on the environmental conditions encountered at these sites.

***Response:*** Thank you for your suggestion. We found that the daytime PBL height in CB and BN during summer can reach 1000-1200 m, while the nocturnal stable boundary layer may shrink to 200-400 m (Wu et al., 2024). Based on this, we have recalculated 72-hour backward trajectories for three characteristic heights (300 m, 500 m, and 1000 m) during typical sampling days to more comprehensively characterize aerosol sources. We corrected the descriptions in Section 3.1 (Page 7, Line 143). We also removed the air mass trajectory in Figure 1 (Page 3) and placed the new air mass trajectory in Figure 2 (Page 7) in the revised MS.

*Reference:*

*Wu Wenlu, Chen Haisha, Guo Jianping, Xu Zhiqi and Zhang Xiaoyan: Regionalization of the Boundary-Layer Height and its Dominant Influence Factors in Summer over China [J]. Chinese Journal of Atmospheric Sciences (in Chinese), 48, 1201-1216, https://doi.org/10.3878/j.issn.1006-9895.2212.22183, 2024.*

- Page 7, Table 1: Maybe illustrate the table with Figure(s) and report the table in the supplementary material file.

  ***Response:*** Following the reviewer's suggestion, we have moved Table 1 to the Supplementary Material (Table S1). The key data from Table 1 have been presented with Figure(s) in the revised MS.

- Lines 149-150: Could you please give a reference of the equations used? Equivalent concentration is obtained by dividing the molar concentration by the charge and not by the ion mass. I have never seen this formula and I assume that the following discussion needs to be revised.

  ***Response:*** We have added the reference: Tian et al., 2018, in the revised MS (see Line 161).

  "Equivalent Concentration $(\mu eq\ m^{-3})$ = $\dfrac{\text{Ion mass concentration } (\mu g\ m-3)}{\text{Molar mass } (g\ mol-1) * \text{Charge number } (z)}$,"

  These formulas are used to calculate the equivalent concentrations of cations and anions in aerosols, with the aim of evaluating the acid-base balance of the aerosol system. The method for calculating equivalent concentration involves dividing the molar concentration of each ion by its charge number (i.e., valence), thereby unifying ions with different charges onto a per-unit-charge basis for comparison. We have given the reference and revised the discussion. See Lines 160-140 in the revised MS.

  *Reference:*

  *Tian, S., Pan, Y., and Wang, Y.: Ion balance and acidity of size-segregated particles during haze episodes in urban Beijing, Atmospheric Research, 201, 159-167, 2018.*

- Lines 163: It would be interesting to treat separately anions and cations.

  *Response:* Following the reviewer's suggestion, we have described the abundances of anions and cations separately in the revised MS (see Lines 173-177).

- Line 166: the authors compare with urban sites… What about the comparison with rural sites?

  *Response:* We added a comparison with rural sites in the revised MS (see Lines 181-183).

- Line 172-179: The authors explain in deep that DMS emitted from phytoplankton can explain the high $SO_4^{2-}$ concentration. If there such a huge marine influence, why $Cl^-$ is so low? Please, reconsider the explanation.

  *Response:* We agree with the reviewer that the contribution from oceanic emissions was insignificant. In fact, such a low concentration of $Cl^-$ is similar to that reported previously by Li et al. (2010) at CB (avg. 0.02 μg m$^{-3}$). Therefore, we modified the interpretation inferring that $SO_4^{2-}$ mainly come from anthropogenic sources in the revised MS (see Lines 194-197).

  *Reference:*

  *Li, L., Wang, W., Feng, J., Zhang, D., Li, H., Gu, Z., Wang, B., Sheng, G., and Fu, J.: Composition, source, mass closure of PM$_{2.5}$ aerosols for four forests in eastern China, Journal of environmental sciences (China), 22, 405-412, https://doi.org/10.1016/S1001-0742(09)60122-4, 2010.*

- Lines 172-179: There is a long discussion on $SO_4^{2-}$, but nothing on $NO_3^-$, which is 73 times higher in winter than in summer at CB. This need an explanation.

  *Response:* We have added explanation in the revised MS (seen Lines 200-207). This could be linked to the increased utilization of coal for domestic heating in winter, which leads to enhanced emissions of gaseous precursors.

- Lines 184-185 and following: I wonder if the two observatories are equipped with $NO_x$, $SO_2$ and $O_3$ analysers. In that case, the discussion would be better supported by experimental measurements of these compounds during aerosol sampling.

  *Response:* Unfortunately both the observatories are not equipped with $NO_x$, $SO_2$ and $O_3$ analyzers, and hence there is no such possibility to provide substantial evidence of gaseous species loading to support the drawn conclusions.

- Figure 4: A concentration of NO3- equal to zero during all the summer for both sites is surprising. I wonder if something went wrong with the analysis. Have you compared this result to those obtained for sites with similar environmental conditions? Pathak et al. (10.5194/acp-9-1711-2009) found, for example, that nitrate is underestimates when the particles collected are deliquescent.

*Response:* The high temperatures in summer might have accelerated the photolytic decomposition of gaseous $NO_2$, reducing its conversion to $NO_3^-$. The CB and BN experience high relative humidity conditions (Fig.2). In particular, BN experiences year-round hot and humid weather, which facilitates efficient removal of $NO_3^-$ aerosols through wet deposition processes. The lower levels of $NO_3^-$ are consistent with those reported by Tanner et al. (2004) for rural (avg. 0.04 µg m$^{-3}$) and background sites (avg. 0.01 µg m$^{-3}$) in the Tennessee Valley, USA, particularly during summer. We included this comparison in the revised MS (see Lines 192-194).

*Reference:*

*Tanner, R. L., Parkhurst, W. J., Valente, M. L., and David Phillips, W.: Regional composition of $PM_{2.5}$ aerosols measured at urban, rural and "background" sites in the Tennessee valley, Atmospheric Environment, 38, 3143-3153, https://doi.org/https://doi.org/10.1016/j.atmosenv.2004.03.023, 2004.*

- Line 251: A correlation with $R^2 = 0.51$ is not so strong

  *Response:* Following the Referee #1 suggestion and considering the influence of the quartz membrane on certain specific ions ($Mg^{2+}$, and $Ca^{2+}$), we removed this part of discussion in the revised MS.

- Line 258: please explain how you calculated nss-$SO_4^{2-}$ and nss-$K^+$ or report a reference.

  *Response:* We have reported a reference in the revised MS (see Line 105).

- Lines 266-267: measurements of $SO_2$, $NO_x$ and $NH_3$ would greatly strengthen the discussion.

  *Response:* We agree with the reviewer's opinion. However, unfortunately, we did not collect real-time measurements of gaseous precursors ($SO_2$, $NO_x$, $NH_3$) synchronously with aerosol sampling.

- Line 271: I have some concern about the concentration of $NO_3^-$ around zero during summer at both sites, thus I'm not confident in the results and discussion about nitrate/sulphate ratios.

  *Response:* As detailed in response to the comment earlier, the high temperatures in summer might have accelerated the photolytic decomposition of gaseous $NO_2$, reducing its conversion to $NO_3^-$. Additionally, the higher precipitation frequency in summer could lead to the washout of $NO_3^-$ from aerosols, further lowering its concentration. Nitrate, particularly ammonium nitrate ($NH_4NO_3$), exhibits strong volatility, and elevated temperatures and humidity during summer may enhance its volatilization, resulting in a decrease in particulate $NO_3^-$ levels. Furthermore, the two forested sites are similar to background sites with minimal influence from industrial activities and traffic emissions.

This is consistent with the concentrations reported by Tanner et al. (2004) for rural (avg. 0.04 µg m⁻³) and background sites (avg. 0.01 µg m⁻³) in the Tennessee Valley, USA, particularly during summer.

In the discussion, we have limited the seasonality of the conclusions, and data from summer (due to extremely low $NO_3^-$ concentrations) were not included in the key conclusions.

*Reference:*

*Tanner, R. L., Parkhurst, W. J., Valente, M. L., and David Phillips, W.: Regional composition of PM₂.₅ aerosols measured at urban, rural and "background" sites in the Tennessee valley, Atmospheric Environment, 38, 3143-3153, https://doi.org/https://doi.org/10.1016/j.atmosenv.2004.03.023, 2004.*

- Line 284: The air masses arriving at CB and BN during summer show a strong marine influence. How do you explain such a low concentration of Cl⁻ (not discussed anywhere else in the article)?

  *Response:* Though the air mass trajectories are showing the transport of oceanic air parcels, the obtained data did not show any such impact at significant level. Since our interpretations mainly depend on the data rather than the air mass trajectories, we did not focus much to discuss it in depth. However, the lower concentration of particulate Cl⁻ in summer may be attributed to the formation of gaseous HCl, which is volatile under high-temperature and high-humidity conditions and could escape from aerosols. The result is comparable to previously reported levels at Mt. Changbai (avg. 0.02 µg m⁻³) and Mt. Dinghu (avg. 0.02 µg m⁻³) (Li et al., 2010). We have added this point in the revised MS (see Lines 270-273).

  *Reference:*

  *Li, L., Wang, W., Feng, J., Zhang, D., Li, H., Gu, Z., Wang, B., Sheng, G., and Fu, J.: Composition, source, mass closure of PM₂.₅ aerosols for four forests in eastern China, Journal of environmental sciences (China), 22, 405-412, https://doi.org/10.1016/S1001-0742(09)60122-4, 2010.*

- Lines 308-311: references are missing about the studies cited.

  *Response:* We have added 3 references: Chow et al. 2007; Schauer eta l., 1999; Chen et al., 2005, in the revised MS (see Lines 320&322).

- General remark: The title promises chemical and stable carbon isotopic compositions, but it is mainly focused on physicochemical characteristic, which is not such a novelty. Only short paragraphs are devoted to stable isotopes, which could really improve the novelty of the work.

  *Response:* We have improved the discussion about stable isotope ratios in the revised MS (see Lines 366-374&384-389). The main objective of this study is to analyze the

physicochemical characteristics of aerosols (e.g., source apportionment, seasonal variations), which serves as the foundation for understanding atmospheric aerosol behavior. Stable carbon isotope ($\delta^{13}C$) analysis is employed as an auxiliary tool to validate the contribution of certain sources.

Thank you again for your detailed and constructive review. We have revised the MS according to all your comments, which improved the quality of the MS. We sincerely hope that these revisions are satisfactory to you for your approval for final publication.

---

## Author Response (AR2)

**Authors' Responses**

***Public justification (visible to the public if the article is accepted and published):***
Thank you for uploading the revised version of the manuscript. I identified that few modifications or clarifications are still required:

Dear Editor,

Thank you very much for your comments and decision on our manuscript for publication with minor revision. The MS is revised accordingly, and our point-by-point responses to all the comments are provided below.

Figure 1: As of today's official country list (e.g., https://worldpopulationreview.com/), Taiwan should not appear in China's map.
***Response:*** According to UN General Assembly Resolution 2758, Taiwan is only a part of China, and the government of the People's Republic of China is the only legitimate government representing China (https://en.wikipedia.org/wiki/United_Nations_General_Assembly_Resolution_2758_(XVI)). However, to avoid misunderstanding to the reader and involving any issue with this map, we removed the Figure 1 in the revised MS.

L177-179: Do not use capital letters in « Ion chromatography » and « Polytetrafluoroethylene ».
***Response:*** We corrected in the revised MS (please see Lines 96-98, referring to the track change acceptance MS. The same below).

Line 182: «The concentrations of all ionic components were corrected for field blanks. ». Please specify how.
***Response:*** The measured concentration of each ion in field blanks was subtracted from the corresponding sample concentrations. We added this point in the revised MS (see Lines 84 & 101-102)

Line 186: «it was found that the blank values ($Na^+$, $Mg^{2+}$, and $Ca^{2+}$) of the blank membranes at BN were relatively high, while the measured concentrations of $Mg^{2+}$, and $Ca^{2+}$ in the samples were very low ». Please specify « relatively high » and « very low ».
***Response:*** We clarified it in the revised MS (see Lines 104-107)

Line 213: please check and rephrase the sentence « Using a UV spectrophotometer to measure the total N's absorbance at 540 nm. »
***Response:*** We have rephrased this sentence in the revised MS (see Line 114).

Line 233 (section 2.3): please check and rephrase the sentence: «Considering the seasonal and diurnal variations in the planetary boundary layer (Wu et al., 2024). »

*Response:* We have rephrased this sentence in the revised MS (see Lines 133-134).

Line 373-375: «The concentrations of main secondary ions were significantly lower compared to those typically observed in urban sites, such as Tianjin, Beijing, Guangzhou, Chongqing in China, Chennai in India, and Hachinohe in Japan (Pathak et al., 2009; Qiao et al., 2019; Pavuluri et al., 2011a; Dong et al., 2023; Sun and Zhang, 2024). However, their concentrations were comparable to or even lower than those reported at rural background sites in France (Bressi et al., 2013), the southeastern United States (Nah et al., 2018), and at the forested site at K-puszta (Kourtchev et al., 2009). ». What is the contradiction here between the 2 sentences ? (Are the « However » and « even lower » justified and correct given the previous sentence ?).

*Response:* We corrected the language error in these sentences in the revised MS (see Lines 175-180).

Line 378: «The concentrations of $SO_4^{2-}$, $NH_4^+$, and $NO_3^-$ in $PM_{2.5}$ during winter were higher than those in summer, being 1.45, 2.55, and 73.00 times, respectively, at CB, and 2.57, 3.76, and 2.25 times, respectively, at BN». Is « being » the correct word ? replace by « by » ?

*Response:* We have replaced "being" with "by" in the revised MS (see Lines 183-184).

Line 681: «The aerosol particles released by plants (primary biological aerosols) exhibit a broader range.» : A range of what ? And broader than what ?

*Response:* The range refers to: "$C_3$ plants emit particles with $\delta^{13}C$ values ranging from -35‰ to -24‰ and $C_4$ plants (e.g., corn, sugarcane) have $\delta^{13}C$ values in the range of -20‰ to -11‰". And this range is broader than the range of fossil fuel combustion (coal, natural gas and petroleum) (-28‰ to -21‰) and $C_3$ plants biomass burning (-28 to -26‰) (Singh et al., 2018) as mentioned in lines 390-392. We included these ranges and relevant text in the revised MS (see Lines 362-364).

Line 701: «resulting in a more positive $\delta^{13}C$ value on that day compared to the regular winter level (-22.8‰ vs. -26.2‰ to -22.1‰)» = is « less negative » meant rather than « more positive»? Otherwise please clarify this sentence.

*Response:* We rephrased "more positive" as " less negative" in the revised MS (see Line 381).